# Reanalysis of the PacIOOS Hawaiian Island Ocean Forecast
# System, an implementation of the Regional Ocean Modeling
# System v3.6
**Dale Partridge[1], Tobias Friedrich[1] and Brian S. Powell[1]**
[1] University of Hawai'i at Mānoa, Department of Oceanography, Marine Sciences Building, 1000 Pope Road, Honolulu,
Hawai'i 96822, USA.

**Abstract**

A 10-year reanalysis of the PacIOOS Hawaiian Island Ocean Forecast System was produced using an incremental strong constraint 4D-Variational data assimilation with the Regional Ocean Modeling System (ROMS v3.6). Observations were assimilated from a range of sources: satellite-derived sea surface temperature (SST), salinity (SSS), and height anomalies (SSHA); depth profiles of temperature and salinity from Argo floats, autonomous SeaGliders, shipboard conductivity-temperature-depth (CTDs); and surface velocity measurements from high frequency radar (HFR). The performance of the state-estimate is examined against a forecast showing an improved representation of the observations, especially the realization of HFR surface currents. EOFs of the increments made during the assimilation to the initial conditions and atmospheric forcing components are computed, revealing the variables that are influential in producing the state-estimate solution and the spatial structure the increments form.

# 1 Introduction

The Pacific Integrated Ocean Observing System [*PacIOOS*, 2018] has produced daily forecasts of the ocean state surrounding the Hawaiian Islands since 2009. To facilitate the forecasts a data assimilation procedure is used to incorporate recent observational data into the model to produce the optimal initial state from which to forecast. A number of modelling studies have been performed with older versions of this model to examine various features of the modeling framework, such as the state estimation [*Matthews et al.*, 2012], nested models [*Janeković et al.*, 2013] and the vorticity budget [*Souza et al.*, 2015]. In this work, we perform an extended reanalysis from 2007 to 2017 in order to produce a consistent data set for further studies of the dynamics around Hawai'i.

The PacIOOS forecast system uses the time-dependent Incremental Strong constraint 4-dimensional Variational Data Assimilation (I4D-Var) scheme [*Courtier et al.*, 1994; *Moore et al.*, 2004] within the Regional Ocean Modeling System (ROMS) [*Moore et al.*, 2011a; *Powell et al.*, 2008; *Matthews et al.*, 2012] to best reduce the residuals between the model and observations, while maintaining a physically consistent solution. The class of methods known as 4D-Var are state-estimation techniques that create a quadratic cost function to be minimized over a defined time window, utilizing observations at the time they occur in a physically consistent manner to adjust the initial state, boundary conditions, and atmospheric forcing to represent the measurements. The I4D-Var scheme is used in operational centers around the world and solves for increments to the model state, boundary conditions, and atmospheric forcing using the model

physics as a constraint. The combination of I4D-Var within ROMS has been used in previous studies of various regions [*Powell et al.*, 2008; *Broquet et al.*, 2009; *Zhang et al.*, 2010; *Matthews et al.*, 2012; *Souza et al.*, 2015]. The details of the model and the observations used within this study are provided in Section 2.

Our model domain covers the Hawaiian Island Archipelago (Figure 1), a dynamically active region for both the ocean and atmosphere. The North Equatorial Current (NEC), flowing from the east, splits upon encountering the island of Hawai'i, with the bulk transport traveling around the south of the island and continuing west, while the North Hawaiian Ridge Current (NHRC) follows the ridge of the other islands in the chain to the north. In the atmosphere, there are persistent trade winds from the northeast that, combined with steep mountainous terrain on the islands, cause wind wakes in lee of the peaks, particularly on the islands of Hawai'i and Maui. This introduces strong temperature gradients, increases the seasonal variability [*Sasaki and Klein*, 2012], and drives currents such as the Hawaiian Lee Countercurrent (HLCC) [*Smith and Grubišić*, 1993; *Xie et al.*, 2001; *Chavanne et al.*, 2002].

There are two main objectives to this study: to assess the skill and performance of the state-estimation model, and to analyze the increments made to the initial, boundary and atmospheric forcing terms. For the first objective, we compare the state-estimate solution with a free-running forecast over the decadal time period and examine how the performance changes over time, utilizing observations derived from satellites and *it situ* measurements. In addition, PacIOOS operates seven high-frequency radar stations sites across the Hawaiian Islands. The first station was constructed in 2010, with the remaining six becoming operational over the period from 2011-2015. These instruments produce high resolution (both spatially and temporally) surface current velocities in the vicinity of the islands of O'ahu and Hawai'i. The use of HFR observations within a state-estimation scheme has been shown to produce a significantly improved representation of surface currents [*Souza et al.*, 2015; *Kerry et al.*, 2016]. The impact of the radar stations will be a key focus point. The performance assessment is achieved through the statistics produced by the state-estimation in Section 3, followed by a comparison with observations in Section 4. The forecast skill, a measure of the accuracy for a forecast system is computed with reference to a persistence assumption (Section 5).

Section 6 focuses on the second objective of the paper, to examine the increments to the initial state and atmospheric forcing to determine how the model is adjusted. By evaluating the Empirical Orthogonal Functions (EOFs) of these increments we determine the spatial patterns in the variability. Since physical modes are not always independent [*Simmons et al.*, 1983],

the interpretation of EOF modes must be undertaken with some caution. As such the result-
ing modes will not necessarily represent a physical phenomenon, but will highlight the vari-
able spatial patterns made over time by the I4D-Var algorithm.

## 2 Numerical Model and Data Assimilation System

### 2.1 Model Configuration

The Regional Ocean Modeling System (ROMS) version 3.6 is used to simulate the phys-
ical ocean around the Hawaiian Islands. ROMS is a free surface, hydrostatic, primitive equa-
tion model using a stretched coordinate system in the vertical to follow the underwater ter-
rain. In order to allow varying time steps for the barotropic and baroclinic components, ROMS
utilizes a split-explicit time stepping scheme (for more details on ROMS, see *Shchepetkin and
McWilliams* [1998, 2003, 2005]).

The Hawaiian Island domain covers 164°W to 153°W longitude and 17°N to 23°N lat-
itude, with bathymetry provided by the Hawaiian Mapping Research Group [*HMRG*, 2017],
shown in Figure 1. The grid has 4km horizontal resolution with 32 vertical s-levels, config-
ured to provide a higher resolution in the more variable upper regions. The configuration model,
including the method for assimilating surface HFRs and the associated vertical stretching scheme,
is identical to the one first presented in *Souza et al.* [2015].

Tidal forcing is produced using the OSU Tidal Prediction Software (OTPS) [*Egbert et al.*,
1994], which is based on the Laplace tidal equations from TOPEX/Poseidon Global Inverse
Solution (TPXO). Tidal constituents included in this simulation are the eight main harmon-
ics; $M_2$, $S_2$, $N_2$, $K_2$, $K_1$, $O_1$, $P_1$, $Q_1$, as well as two long period and one non-linear constituent;
$M_f$, $M_m$ and $M_4$. To avoid any long term drifting of the tidal phases related to constituents we
do not consider, the tidal harmonics are updated each year to define the phases in terms of the
start of that year.

Lateral boundary conditions are taken from the HYbrid Coordinate Ocean Model (HY-
COM) [*Chassignet et al.*, 2007] and are applied daily. Within ROMs, we apply the boundary
differently for each variable; Chapman [*Chapman*, 1985] conditions are applied to the free sur-
face, Flather [*Flather*, 1976] conditions for transferring momentum from 2D barotropic en-
ergy out of the domain, while the 3D momentum and tracers variables are clamped to match
HYCOM. A sponge layer of 12 grid cells (48km) linearly relaxes the viscosity by a factor of
four and diffusivity by a factor of two close to the boundary to account for imbalances between
HYCOM and ROMS.

From 2007-2009, atmospheric forcing fields (excluding the wind), are provided by the National Center for Environmental Prediction (NCEP) reanalysis fields [*Kistler et al.*, 2001]. For the wind forcing, a combination of two different forcings is utilized: i) a 1/2° resolution CORA/NCEP wind product [*Milliff et al.*, 2004] that combines QuikScat measurements with NCEP wind fields; and, (ii) The CORA/NCEP winds blended with the results from a 1/12° resolution PSU/NCAR mesoscale model (MM5; *Yang et al.* [2008a]) of the Hawaiian islands [*Van Nguyen et al.*, 2010]. The MM5 model was forced at its boundaries with the global NCEP fields; hence, it is a consistent dynamical downscaling of the global fields. The MM5 model domain is smaller than the ocean grid domain, extending only to 160.5°W in the lee. Therefore, for (ii), we must blend the modeled and CORA/NCEP winds to generate a consistent field for the entire region with 1/12° winds where available and 1/2° winds everywhere else.

To blend the two, we convert the MM5 winds to anomalies by subtracting a 30 day mean centered about the record of interest. We compute the mean for the same period from the CORA/NCEP winds. The difference between the two means provides a bias estimate. The bias is removed from the MM5 anomalies and the CORA/NCEP mean is added. Within a 1° box around the boundary of the MM5 data, we taper the anomalies to zero with a cosine filter to avoid abrupt changes to the field. This step ensures that the mean of the CORA/NCEP field is preserved while its structure and variability is greatly enhanced by the MM5.

From July 2009, atmospheric forcing is provided locally by a high-resolution Weather Regional Forecast (WRF) model [*WRF-ARW*, 2017]. WRF supplies information about surface air pressure, surface air temperature, long- and short-wave radiation, relative humidity, rain fall rate, and 10m wind speeds. The ocean model is forced using this data every six hours, taken from the atmospheric model with 6km resolution across the entire domain.

Prior to the experiment, a six-year non-assimilative model was run using the same initial state, boundary conditions, and atmospheric forcing. The variability of the model is used to produce an estimate of the background error covariances used within I4D-Var, as well as the mean sea surface height to use with sea level anomaly observations.

The cost function of the I4D-Var method penalizes for the increments made to the initial conditions, the boundary conditions and the forcing; and for the deviations of the model state from the observations. A detailed derivation of the cost function can be found in [*Kerry et al.*, 2016; *Penenko*, 2009; *Weaver et al.*, 2003; *Stammer et al.*, 2002; *Talagrand and Courtier*, 1987]. To formulate the solution, we must provide estimates of the uncertainty matrices in both the model and observations. The model uncertainty matrix, $\mathbf{P}$, is estimated using the variabil-

ity of the six-year run described above, while observation uncertainty matrix, **R**, is assumed
to be diagonal, (i.e. observations are independent). The implementation of I4D-Var in ROMS
is covered extensively in [*Moore et al.*, 2011a,b,c].

## 2.2 Experiment Setup

The reanalysis covers a period of 10 years, from July 2007 to July 2017. The period of
assimilation for the I4D-Var cycles is four days, which corresponds to the limit of the linear-
ity assumption within the domain [*Matthews et al.*, 2011]. The atmospheric forcing is adjusted
every six hours, while the boundaries are every $12h$. An analysis of these adjustments is per-
formed in Section 6.
During each I4D-var cycle, a minimization procedure is applied. The non-linear model
is first integrated forward to estimate the background state (the first *outer* loop). Then the tangent-
linear and adjoint models are integrated in multiple *inner* loops to minimise the cost function
($J$). After the last inner loop the non-linear model is updated (see Figure 1 of *Moore et al.* [2011a]).
Prior methodological experiments yielded that for our setting a sufficient reduction in $J$ (and
an acceptable computational cost) can be achieved using a single outer loop with 13 inner loops
[*Souza et al.*, 2015].
Four and eight day forecasts are performed from the end of each cycle using the assim-
ilated state as initial conditions, and the short-range (1-4 days) and mid-range (5-8 days) fore-
casts are evaluated for skill.

## 2.3 Observations

Observational data used within this study include satellite measurements of the ocean
surface of temperature, height, and salinity, *in situ* depth profiles of temperature and salinity,
and surface HFR velocities from High Frequency Radar. Observations within one Rossby ra-
dius ($\sim$80 km) of the domain's boundary are neglected. It should be emphasized that no ob-
servations were withheld from the assimilation for the purpose of validation. The I4D-Var method
seeks to represent the observations by exploiting the linearized model dynamics. Therefor, all
available observations are used to constrain this representation.
### *2.3.1 Satellite Derived Measurements*

Sea Surface Temperature (SST) observations are available from two sources at differ-

ent time periods: initially we used the Global Ocean Data Assimilation Experiment High Res-
olution Sea Surface Temperature (GHRSST) Level 4 OSTIA Global Foundation Sea Surface
Temperature Analysis [*PO.DAAC*, 2005], referred to as OSTIA for this work. The data are dis-
tributed by the Physical Oceanography Distributed Active Archive Center (PO.DAAC), using
optimal interpolation to combine data from the Advanced Very High Resolution Radiometer
(AVHRR),the Advanced Along Track Scanning Radiometer (AATSR), the Spinning Enhanced
Visible and Infrared Imager (SEVIRI), the Advanced Microwave Scanning Radiometer-EOS
(AMSRE), the Tropical Rainfall Measuring Mission Microwave Imager (TMI), and *in situ* data.
This distribution provides a highly smoothed daily gridded global dataset at the surface at a
6km spatial resolution, accurate between $0.2 - 0.5\,^\circ$C in the domain.

Beginning in April 2008, we switched to using the GHRSST Level 4 K10_SST Global

1 meter Sea Surface Temperature Analysis data set [*PO.DAAC*, 2008], produced by the Naval
Oceanographic Office, and is referred to as NAVO for this work. Also distributed by PO.DAAC,
this product combines, in a weighted average, data from AVHRR, AMSRE and the Geosta-
tionary Operational Environmental Satellite (GOES) Imager. This distribution provides a daily
gridded global dataset at 1 meter depth at a 10km spatial resolution, accurate to $0.4\,^\circ$C in the
domain.

Sea Surface Height (SSH) observations are derived using sea level anomaly data from

the Archiving, Validation and Interpretation of Satellite Oceanographic data (AVISO) delayed
time along track information. The data comes from multiple altimeter satellites measuring the
anomaly with respect to a twenty-year mean SSH, homogenized against one of the missions
to ensure consistency. Each track has approximately 7km spatial resolution and will usually
make multiple passes through our domain each day. To convert from sea level anomaly to sea
surface height we add the mean SSH field taken from the six-year model run described above,
to which we add the barotropic tidal prediction from TPXO. The accuracy of the swaths de-
pend on the source satellite and ranges from $5-7$ cm. We use the AVISO product that has
been fully filtered and quality controlled until May 2016. At the time of the experiment, the
delayed time data were unavailable beyond May 2016, so the near real-time data were used.

Sea Surface Salinity (SSS) data are taken from Aquarius missions daily L3 gridded data

set [*PO.DAAC*, 2015] distributed by PO.DAAC. The satellite uses a combination of radiome-
ters and scatterometers to estimate the surface salinity, mapped to a coarse $1^\circ$ resolution. Er-
rors for this product are around 0.2 ppt. Data for this product are available from August 2011
until June 2015.

### 2.3.2 In Situ *Measurements*

Depth profiles of temperature and salinity are obtained from threes sources: the Hawai'i
Ocean Time-Series (HOT) shipboard Conductivity Temperature Depth (CTD) casts, the global
network of Argo floats, and autonomous SeaGliders operated by the University of Hawai'i.
The HOT project conducts monthly cruises to the deep water station *A Long-term Olig-*
*otrophic Habitat Assessment* (ALOHA) (located at 23° 45'N, 158° 00'W, see Figure 1) in or-
der to develop continuous data sets of physical and biochemical ocean parameters. CTD sta-
tions of temperature and salinity are concentrated in the region around the station; although
some are also established along the ship route.
HOT also conducts regular SeaGlider missions departing from station ALOHA. In ad-
dition, PacIOOS conducts occasional SeaGlider surveys in areas close to the south coast of
O'ahu. The buoyancy driven autonomous underwater vehicles take profiles and transects at depth
of temperature and salinity.
Observations from the global Argo float network are available from the Argo array Net-
work [*USGODAE*, 2016]. The free-drifting floats profile temperature and salinity during as-
cension and descension every 10 days of depths down to 2000m [*Oka and Ando*, 2004]. Argo
measurements tend to occur in the model domain at a rate of about 1-2 profiles per day.
Representational errors for HOT CTDs, Argo Floats, and SeaGliders are defined by the
variance of observational data from all available sources across our domain sorted into depth
bins. These profiles resemble a typical temperature/salinity profile, with a peak temperature
error of 0.8 K, and peak salinity error of 0.15 ppt occurring in the mix layer at a depth around
100m.

### 2.3.3 *High Frequency Radar Measurements*

HFR measurements of surface currents are available from PacIOOS at seven sites around
the Hawaiian islands: five around the south-west of O'ahu and two on the east coast of the
Hawai'i. Data are available from the first site in October, 2010 with the other sites coming on-
line at various times, the most recent being October, 2015. The range for the HFRs on O'ahu
extend approximately 150km from the coast, while the two Hawai'i sites are focused on cur-
rents around the Northeast of the island and have a shorter range. At the range limits, HFR
data are less reliable due to the higher noise level of the returns. Figure 2 shows the percent-
age availability of data in the region. HFR measurements from any return location that it miss-
ing more than 20% of its data over the 4-day assimilation period are ignored. Both spatially
and temporally, the resolution for all sites is significantly higher than the model resolution. The
HFR data are low-pass filtered at 3 hours to remove the high frequency signals that may not
be resolved by the model (atmospheric forcing fields are every 3 hours). We then provide the
spatial field of data every 3 hours. The associated error is calculated individually for each spa-
tial point as the accuracy of the measurements is determined by the levels of interference, which
increases with range. For each observation point we calculate the power spectral density and
calculate the noise as per *Zanife et al.* [2003], with a minimum of 7 cm/s. At the extreme, er-
rors may reach 17 cm/s.

The number of observations for each four day cycle from all sources are shown in Fig-

ure 3. Sea surface temperature measurements from both OSTIA and NAVO are consistently
the most available observation source, and by the end of the time period HFR is supplying a
similar quantity. *In situ* measurements, which include both temperature and salinity for each
of the instruments, provide a smaller amount of data by an order of magnitude.

## 3 Assimilation Statistics

In this section we examine the state estimate to quantify the performance during our time

period.

### 3.1 Cost Function Reduction

I4D-Var minimizes the residuals between the model and observations over each 4-day

cycle. We calculate the percentage reduction between the initial and final cost function for each
cycle to assess how the assimilation performs over time. Additionally, the I4D-Var algorithm
reports the individual contributions by the state variables considered by the data assimation
to the total cost function. Hence we can examine the cost function in detail for those obser-
vation types that are most critical for its reduction. However, it should be noted that for this
decomposition we do not distinguish between observation sources.

Figure 4 shows the time series of the total reduction and the percentage reduction in the

cost function for each of the variables we observe: sea surface height, temperature, salinity
and HFR. A value of 0 means the final cost function is the same as the initial and no reduc-
tion has occurred. The plot is split into two distinct time periods, before and after the HFR

observations in order to assess changes in the relative contributions of each variable to the over-

all reduction.

The total cost function of all data (Figure 4A) is – on average – halved for each cycle, with an improvement from 49% of the original value to 55% when HFR observations are available. Looking at the breakdown in Figure 4B-E, we see that the final cost function associated with the other observed variables: sea surface height, temperature, and salinity, is reduced by a smaller percentage than before HFR was included. Given that the structure of the cost function is determined by the type and number of observations, this change in contribution to the cost function reduction can be expected when adding a large number of HFR measurements to the data assimilation.

Salinity measurements tend to contribute the least improvement ,ranging from 34% (pre-HFR) to 16% (post-HFR). Salinity data are least numerous (Figure 3) and SSS fields taken from Aguarius are subject to high noise levels (0.2 ppt) and coarse spatial resolution. The mid-2014 drop in cost function reduction for salinity data coincides with the loss of two SeaGliders. After the cessation of SeaGlider missions salinity data were only available through Aquarius (until mid 2015) and sporadic Argo profiles.

The cost function associated with HFR measurements is reduced by 60% of the initial value, meaning the model is closer to the HFR observations after the assimilation.

### 3.2 Optimality

Another measure of the performance is the theoretical minimum value of the cost function ($J_{min}$). For a linear system and assuming that the error matrices $\mathbf{P}$ and $\mathbf{R}$ have been determined correctly, $J_{min}$ is a chi-squared variable whose degrees of freedom are given by the number of assimilated observations ($N_{obs}$) [*Bennett*, 2002]. The expected value of $J_{min}$ is then given by:

$$< J_{min} >= \frac{N_{obs}}{2},$$ (1)

Using above equation, an optimality value ($\gamma$) can be defined:

$$\gamma = \frac{2 \cdot J_{min}}{N_{obs}},$$ (2)

which should reach a value of 1 with a standard deviation of $\sqrt{2/N_{obs}}$.
This optimality value provides a simple representation of how consistently the error ma-
trices ($\mathbf{P}$ and $\mathbf{R}$) are specified, since the error covariances normalize the cost function. Fig-
ure 5 shows a time-series of the calculated optimality value for the model run, in addition to
a timeline of the availability of certain observations for reference. Over the full time period
the mean optimality is $0.95$. However, there are large significant deviations over the course
of the time period. In the pre-HFR period the optimality is low, suggesting that the error bounds
on observations are too wide. Since SST is the dominant source of observations before HFR,
the prescribed errors associated with SST may be too large.
Post-HFR, the optimality value increases, suggesting the errors in this period are under-
estimated. A large optimality value arises when the cost function is large (*i.e.* large differences
between the model and observations). There were two anomalous cycles in 2011, the first co-
incides with the introduction of a second radar site. From 2012 onwards the optimality value
is generally good, if highly variable. The increase in optimality given the available observa-
tions points to an underestimation of HFR errors, or at the least a persistent difference between
the model and HFR observations.
### 3.3 Error Consistency
The consistency of the assimilation can be assessed by comparing the error matrices $\mathbf{P}$
and $\mathbf{R}$ specified *a priori* with the observation and background error covariances determined
*a posteriori* [*Desroziers et al.*, 2005]. Using the difference between the observation $j$ ($\mathbf{y}_j$) and
the modeled background value ($\mathbf{x}^b$) mapped to the observation location by the operator $\mathcal{H}_j$:

$$d_j^{ob} = y_j - \mathcal{H}_j(x^b), \tag{3}$$

and the difference between $\mathbf{x}^b$ and analysis value ($\mathbf{x}^a$) mapped to the observation loca-
tion:

$$d_j^{ab} = \mathcal{H}_j(x^a) - \mathcal{H}_j(x^b), \tag{4}$$

one can compute the expected *a posteriori* background error:

$$\widetilde{(\sigma_i^b)}^2 = \frac{1}{p_i} \sum_{j=1}^{p_i} (\mathcal{H}_j(x^a) - \mathcal{H}_j(x^b))(y_j - \mathcal{H}_j(x^b)), \tag{5}$$

where $i$ refers to the observation type and $p_i$ is the number of observations of that type.
Similarly, using the difference between the the observation $j$ and the modeled analysis
value ($\mathrm{x}^a$) mapped to the observation:

$$d_j^{oa} = y_j - \mathcal{H}_j(x^a), \tag{6}$$

the expected *a posteriori* observation error can be calculated:

$$\widetilde{(\sigma_i^b)}^2 = \frac{1}{p_i} \sum_{j=1}^{p_i} (y_j - \mathcal{H}_j(x^a))(y_j - \mathcal{H}_j(x^b)). \tag{7}$$

For a detailed description of above dignostics the reader is referred to *Desroziers et al.*
[2005, 2009]. If the variances in $\mathbf{P}$ and $\mathbf{R}$ are correctly specified *a priori*, they will be con-
sistent with the *a posteriori* errors defined above. Figure 6 shows both the *a priori* and *a pos-
*teriori* errors for the remotely sensed data. The observation *a priori* values are calculated as
the mean error of the observations in each cycle, while the background *a priori* values are de-
fined as the variability of a free running non-linear model. If the *a posteriori* errors are typ-
ically larger then the *a priori*, it implies the initial errors in $\mathbf{P}$ and $\mathbf{R}$ are underestimated. Con-
versely, if they are smaller the initial errors are likely overestimated.
Figure 6A shows that sea surface height errors are consistent, while sea surface temper-
ature, Figure 6B suggests the *a priori* errors are overestimated. The *a priori* observation er-
rors for NAVO SST observations are defined with a minimum error of $0.4$ K, but the *a pos-
*teriori* are more typically around $0.25$ K. The *a priori* background errors also also appear over-
estimated.
Sea surface salinity observation errors (fig. 6C) are slightly underestimated but gener-
ally consistent, as are the background errors. The HFR observation errors (fig. 6D) also ap-
pear to be underestimated, with most *a priori* errors close to the minimum value of 7 cm/s.
The *a posteriori* errors suggest a typical value of around $12-15$ cm/s would be more appro-
priate. The *a priori* background errors are reasonably consistent with the *a posteriori*, if any-
thing they are slightly overestimated.
This error consistency analysis supports the conclusions in Section 3.2 that the SST ob-
servation errors are overestimated and HFR values are underestimated. It is worth noting that
these diagnostics are only estimates used to characterize the errors and are not the true pos-
terior error.

## 4 Comparison with Observations

Because I4D-Var relies on the model physics to represent observations through time, it
should provide better forecasts. Time-invariant methods (3D-Var, Optimal Interpolation) that
perturb the state at single times may better reduce the time-fixed cost function, but can add
non-physical structures that generate noisy forecasts.
In this section, we examine the state estimate solution by comparing the model to ob-
servations. For reference, the observations are also compared against the forecast starting from
the same time as each state-estimate cycle. The initial and boundary as well as atmospheric
and tidal forcings are initially the same for both runs; however, the initial and boundary con-
ditions and atmospheric forcing are altered as part of the state estimate solution.
For comparing fields we use the Root Mean Squared Anomaly (RMSA) and the Anomaly
Correlation Coefficient (ACC), defined as:

$$\text{RMSA}(\mathbf{x}, \mathbf{y}) = \sqrt{\frac{1}{N} \sum_{i=1}^{N} \left( (x_i - \bar{x}) - (y_i - \bar{y}) \right)^2} \tag{8}$$

$$\text{and} \qquad \text{ACC}(\mathbf{x}, \mathbf{y}) = \frac{\sum_{i=1}^{N} (x_i - \bar{x})(y_i - \bar{y})}{\sqrt{\sum_{i=1}^{N} (x_i - \bar{x})^2 \sum (y_i - \bar{y})^2}}, \tag{9}$$

where $N$ is the number of observations and $x$ are the model values at the same loca-
tion and time as the observations $y$. The RMSA provides a measure of the residual between
the model and observations, while the ACC determines the strength of the relationship between
the two. We can calculate values for a single spatial point throughout time, or for all spatial
points at a single time; however, we require there must be at least 20 observation values avail-
able to get a representative statistic. The gridded satellite products are ideally suited to this
analysis, while the depth profiles from *in situ* measurements are binned into 50 m depth lay-
ers to ensure a minimum number of values. Here it must be noted that our validation is lim-
ited to data that have been employed for the assimilation. The I4D-Var scheme uses the lin-
earized model dynamics to produce the covariance between the model and the observations.
This allows the model to optimally represent the observations in time and space rather than
replicate them. As such, the desire is to use every available observation to constrain this rep-
resentation. Unlike time-invariant statistical methods, we do not withold any observations be-
cause they are sampling the dynamical sub-spaces of a system of unknown dimension. Since
the observations covary in space and time, some particular observations may not have a sig-
nificant impact on the cost function and their representation may suffer. We seek to identify
these results.

### 4.1 Remotely Sensed Observations

Figure 7 shows the RMSA between the observations and the models for each source of
remotely observed data. The state-estimate solution reduces the RMSA compared with the fore-
cast by 1.58 cm (17%), 0.07 K (24%), 0.01 ppt (3%) and 8.39 cm/s (37%) for sea surface height,
sea surface temperature, sea surface salinity and HFR respectively. In Figure 7A the RMSA
of the state-estimate solution is close to the typical observational error of 7 cm, while in Fig-
ure 7B we see the RMSA is comfortably less than the 0.4 K representative error. Sea surface
salinity is only marginally improved by the state-estimate solution, but is slightly over the pre-
scribed observational error of 0.2 ppt. The RMSA of the currents associated with HFR ob-
servations, shown in Figure 7D, is improved greatly by the state-estimation; however, the mean
value of 14 cm is around double the typical error prescribed *a priori* of 7 cm. As shown in
the previous sections, this error was underestimated.
The ACC is also improved by the state-estimate for all variables, as shown in Figure 8.
For sea surface height, temperature and salinity the improvement is small due to a significant
agreement in the forecast with gains of 0.03, 0.02, and 0.01 respectively. The improvement
in HFR is much more significant, with an average correlation improvement from 0.35 to 0.68.
As shown in Figure 8D the free-running forecast model can diverge from the observations enough
to become negatively correlated over a cycle, while the state-estimate solution is consistently
positively correlated.
Figure 9 shows the spatial RMSA between the forecast and analyses model solutions and
the observations for both sources of sea surface temperature observations: OSTIA and NAVO.
In both cases there is a clear reduction in the RMSA, with the largest source of error in the
areas leeward of the islands, most notably the island of Hawai'i. This is due to higher heat
flux variability from a reduction in cloud cover [*Yang et al.*, 2008b; *Matthews et al.*, 2012]. Even
in this peak area, the state-estimate solution is around the observational error of representa-
tiveness of 0.4 K, meaning the model is performing well with regards to SST.

Both RMSA and ACC between the experiments and HFR observations are shown in Figure 10 for the island of O'ahu. The RMSA of the free-running forecast is reasonably uniform across the region covered by the HFR, around $20-25$ cm/s with some varying values around the extent of the radar coverage. The inclusion of HFR observations in the state-estimate solution leads to significantly reduced values of $12-15$ cm/s, a reduction of almost half. The ACC is also significantly improved from a weak correlation to a consistently strong positive one.

As discussed in *Souza et al.* [2015], there are several reasons the model can differ from surface current observations: the discretization of the model, imperfect stratification, differing barotropic-to-baroclinic tide conversion at Kaena ridge, or mixing parameters that do not capture the real baroclinic mixing. This may lead to a different location of the currents in the model from those observed by the HFR; however, the model does a good job reducing these errors [*Janeković and Powell*, 2012]. The HFRs located on the island of Hawai'i have a smaller coverage region, but the level of improvement from the forecast to the state-estimate solution is consistent with the O'ahu results shown here.

### 4.2 Subsurface Observations

The *in situ* observation sources: Argo floats, Seagliders and HOT CTDs also show an improvement in the state estimate over the forecast. The subsurface temperature RMSA values are reduced by an average of $0.03$ K and salinity by $0.01$ ppt. The average RMSA is within the representative errors for both variables, $0.8$ K and $0.15$ ppt, respectively. However, there are several occasions when the RMSA value for a cycle exceeds that limit when there are very few *in situ* observations available.

Figure 11 shows the RMSA and ACC profiles for temperature and salinity respectively for each source of subsurface observation. For all three sources, the greatest RMSA between the models and observations is along the thermocline where minor differences in thermocline depth leads to temperature differences. The state-estimate improves the RMSA in this region by $10-15$ %. The thermocline location is also the source of lowest correlation between the observations and the model, which is improved by the state-estimate by $\sim 5$ %. There is a high RMSA for SeaGliders at the base of their profiles (close to $1000$ m). In this instance the state-estimate does not result in an improvement of the forecast. Many of the Glider missions operated in the shallow waters off the south coast of O'ahu where processes are at much finer

scale than can be resolved at 4 km resolution. As such, the observational representation er-
rors were higher.
For subsurface salinity (fig. 11, lower panel), the improvements made by the state-estimate
solution occur almost exclusively above 500 m for Argo floats and HOT CTDs. As with tem-
perature the largest improvement is at the top of the thermocline. There are some low ACC
values lower down in the profile between both models and the observations, but both the fore-
cast and state-estimate perform equally at this depth. SeaGliders produce the biggest improve-
ment in subsurface salinity model performance, with the state-estimate solution up to 20 %
better than the forecast for both RMSA and ACC. The peak improvement is at the top of the
thermocline, but there are improvements throughout the profile.

## 5 Forecast Skill

In this section we quantify the model skill by using a skill score, evaluated as the im-
provement against a reference field [*Murphy*, 1988]. For the reference, we take the model value
at the spatial location of each observation at the time of initialization for each 8-day cycle and
assume persistence of this value throughout the 8-day cycle (persistence assumption). The skill
score (SS) for the state estimate analysis and forecast are then defined using the ratios of RM-
SAs with respect to the observations:

$$SS_a = 1 - \frac{\mathrm{RMSA}(\mathbf{x}^a, \mathbf{y})}{\mathrm{RMSA}(\mathbf{x}^0, \mathbf{y})}, \qquad (10)$$

$$SS_f = 1 - \frac{\mathrm{RMSA}(\mathbf{x}^f, \mathbf{y})}{\mathrm{RMSA}(\mathbf{x}^0, \mathbf{y})}, \qquad (11)$$

where the superscripts $a$, $f$, and 0 refer to the analysis, free-running forecast and per-
sistence, respectively; and $y$ indicates the observations. Under this measure, a SS of 1 repre-
sents a perfect fit between the model and observations, while a value of zero indicates where
the model and persistence values perform exactly the same. If the model is better than per-
sistence, then the skill score will lie in the range $0 < SS < 1$ and the degree of improve-
ment over persistence is determined by how close to 1 the score is. Conversely, a negative SS
means the model is further from the observations than persistence.
For this verification we wish to examine the effect of forecast length on the skill. Start-
ing with the same initial conditions as each state estimate cycle we produce an eight day fore-
cast, the length of two state estimate cycles. The RMSA is calculated every 3 hours for each
8-day forecast, the corresponding state-estimate cycles, and the persistence field from the start
of the forecast.

Figure 12 shows the mean SS over all cycles for remotely sensed observations. For SSH,

SST and HFR, the skill for both the state-estimation and free-running forecast is positive through-
out, indicating that both models are successful over persistence in representing those variables.
SSS however is close to zero and slightly negative meaning the models provide no better in-
formation than persistence. SST values are consistently the highest, with a reduction in skill
versus persistence for both models once per day. This is expected as initial conditions are used
for persistence values and the diurnal cycle will move ocean temperatures close to this per-
sistence value once per day. The state-estimate skill for HFR has a consistent value of $0.5$ re-
gardless of the forecast day, while the skill of the free-running forecast decreases within the
first 12 hours and is closer to $0.2$ for the rest of the forecast period. This decrease in skill is
driven by the fact that the radials are dominated by the semi-diurnal baroclinic and barotropic
tides.

## 6 Analysis of Increments

During each I4D-Var 4-day window, the initial model field, as well as time-varying bound-

ary and surface forcings are adjusted to minimize the residuals. The initial condition incre-
ments form a single record for each cycle, while the boundary and surface forcings are per-
turbed every time they are applied to the model. The perturbations applied to the boundary
exhibit only a minor influence on the model (not shown), due to the mean advection speed ($\approx$
20 cm $s^{-1}$) and sponge layer dampening near the boundaries. We focus our analysis on the
increments of the initial conditions and the surface forcing.

Because we are analyzing the increments (rather than the state) to the initial conditions

and forcing fields, the mean increment should be zero (unless there is a bias in the model),
and we are looking to examine the variability. Over the entire reanalysis period, the mean bias
between the model and observations for the different types are: temperature (-0.0048 K), salin-
ity (0.0049 ppt), SSH (-7 mm), and HFR (0.06 cm $s^{-1}$). A consistent pattern or principal com-
ponent may suggest a repeated correction to account for missing or mis-represented physics
in the model.

Over the 10 year reanalysis, there are 917 analysis cycles with sixteen surface forcing

adjustments (four per day) per cycle. We calculated the Empirical Orthogonal Functions (EOFs)
[*Hannachi*, 2004] of the increments applied to the forcing and the initial conditions to ana-
lyze the dominant spatial patterns of the adjustments.
For each cycle, the initial perturbation of the primary model prognostic variables are ex-
amined: sea surface height, temperature, salinity, east-west velocity and north-south velocity.
With the exception of sea surface height, each variable is averaged over the upper 100 m to
cover the mixed layer depth in the domain [*Matthews et al.*, 2012]. The increments for salin-
ity and sea surface height as a percentage of the initial conditions are insignificant ($< 1\%$),
while temperature increments ($2 - 10\%$) and the two velocity fields ($10 - 20\%$) are signif-
icant enough to analyze.
The assimilation was configured to optimize the surface forcing increments every 6 hours
(to avoid over-adjustment). The time of day potentially impacts forcing variables, particularly
surface heat flux, so we calculate EOFs on the increments for each of the four distinct times
of day they occur (00, 06, 12, 18 UTC). Due to the size of the model grid, the number of records
and the computational resources available the EOF calculation is limited to a 4-year period,
approximately 1500 records. Several different periods were examined with no significant dif-
ferences in the structure of the modes or their percentage variance explained. The time of day
does impact the percentage variance explained by each mode, most notably for surface heat
flux where the effect of diurnal solar heating occurs. However, the overall locations and mag-
nitudes of the peaks/troughs as well as the temporal evolution of PCs do not exhibit signif-
icant differences for each time of day, so we present one of the modes for each considered vari-
able.
The four key surface forcing terms are: surface heat flux, surface salinity flux, east-west
wind stress, and north-south wind stress. Of these, increments in surface salinity flux are quite
small compared to their initial value, while increments in surface heat flux ($10 - 15\%$ of ini-
tial value) and the wind stresses ($15 - 20\%$ of initial value) are significant.
For surface heat flux and near surface temperature, we observe that the EOF1 modes rep-
resent $63\%$ and $20.8\%$ of the variability respectively with a consistent sign over the region (Fig-
ure 13). This mode essentially accounts for the bias between our ocean model and the WRF
atmospheric model used to force the surface. Unfortunately, WRF was not integrated loosely
coupled to the ROMS using the ROMS SST field, rather it was run using persistent estimates
of daily SST during its integration. It must be noted, however, that the monopole structure of
the EOF1 does not represent a constant offset between ROMS and WRF since the actual per-
turbation of surface heatflux and increment applied to near-surface temperature are given by

the products of the respective EOF1 and the PC1. As can be seen in the lower panel of Figure 13, the temporal evolution of the PC1 for both surface heatflux and near-surface temperature adjustments is dominated by high-frequency, non-physical variance.

The EOF1 modes of the near-surface velocity increments explain 26.1% and 20.8% of the variance respectively. Both modes exhibit a strong impact south of the main Hawaiian Islands. The structure of the wind stress curl in this region results in the spin-up of cyclonic and anticyclonic eddies to the north and south side of the lee side of each island respectively [*Chavanne et al.*, 2002]. As a consequence, a zone of strong current shear is created between the North Equatorial Current and the Hawaiian Lee Counter Current [*Lumpkin and Flament*, 2013] (see also Figure 1). The EOF1 modes of the near-surface velocity increments are responsible for adjusting the state estimate for the significant eddy activity in the lee of Hawaiʻi.

The EOFs of surface wind stress increments are confined to relatively small regions of the model domain (Figures 14 and 15). A significant change occurs after the HFR observations come online. During the period prior to the availability of the HFR data (June, 2007– September, 2010), the wind stress was primarily adjusted in the lee regions where the winds are forced between island (*e.g.,* Kaiwi and ʻAlenuihāhā Channels and to a smaller degree over the the Kauaʻi Channel, Figure 14). The wind stress curl in these regions plays an important role as a vorticity source to the ocean [*Souza et al.*, 2015]. Hence adjustment of wind stress in the channels between the islands is critical for a reliable representation of ocean conditions. The magnitude and sign of PCs of the wind stress adjustments for this period are driven by day-to-day variability (Figure 14, lower panels). Also, the PCs of the East-West wind stress and North-South wind stress are largely uncorrelated aggravating an interpretation of the adjustments in terms of a larger scale atmospheric pattern or wind stress curl.

With the integration of the HFR measurements (October 2010), the dominant wind stress increments occur across the shallow region close to the south coast of Oʻahu (Figure 15). The first mode for both East-West and North-South wind stress exhibits a monopole structure adjusting the wind stress uniformly across the area covered by the HFR and its vicinity. The second modes have an east-west dipole structure that will either increase or decrease the wind stress shear around the HFR region. Similarly to the pre-HFR period, the PCs of the wind stress increments are dominated by day-to-day variability and do not represent a physical mode.

## 7 Summary

We have presented a 10-year reanalysis of the PacIOOS Hawaiian Island Ocean Forecast System and assessed the performance of the state-estimate solution and free-running forecasts. Using a time-dependent Incremental Strong constraint 4-dimensional Variational Data Assimilation (I4D-Var) scheme, we show that the model represents the observational data well over the time period. The state-estimate solution reduces the RMSA compared to the forecast by 3% (salinity) to 37% (surface velocities). A limitation of the model-observation comparison is given by the fact that – in the absence of a sufficient number of independent observations – only assimilated data could be used for the validation.

The largest reduction of the cost function of the state-estimate solution occurs when minimizing the residuals to HFR data, with SST also accounting for a significant improvement. On average, the assimilation achieves the near-optimal solution; however, the variability is heavily influenced by the HFR observations. The analysis suggests that the observational errors associated with HFR are too low and results could be improved by redefining these errors. This is supported by the increase in variability and upward trend of optimality towards the end of the time period where HFR observations are most numerous.

The increments made by the reanalysis have revealed that sea surface height and salinity initial conditions are not significantly adjusted by the I4D-Var procedure; whereas temperature and velocity account for a significant change from the forecast field. For the atmospheric forcing, surface salinity is insignificant, but the adjustments made surface heat flux and wind stresses alter the forcings by up to 20%. This corresponds to cost function statistics that point to HFR and temperature as the two dominant observation sources.

The dominant EOF mode for adjustments of surface heat flux and near-surface temperature exhibit a monopole structure indicating a slight bias correction between the ocean and atmospheric model. The leading modes of wind stress increments are concentrated in the region south of Oʻahu. The wind stress heavily influences the surface currents and adjustments are mostly made as a consequence to HFR data. Additional analysis reveals that wind stress adjustments in the channels between the islands dominated the increments in the period prior to the radar-based measurements of surface currents.

The reanalysis has provided the testing for improvements to the PacIOOS operational forecast system. The data are being used to update the back catalog available to the public at www.pacioos.hawaii.edu and will influence the future results from daily forecasts. Analysis

of the I4D-Var increments has provided a greater understanding of the variability in the re-
gion and will provide the basis for a move towards ensemble forecasting in the region.

## 8  Code and Data Availability

The latest ROMS code for running the model is available as an open source software
package distributed freely from http://www.myroms.org. The python code for working with
the output is available from github.com/powellb/seapy.
Model initial conditions and boundary forcing comes from the HYbrid Coordinate Ocean
Model (hycom.org). Atmospheric surface forcing and HFRadar observations are distributed
through the PacIOOS data portal (pacioos.hawaii.edu).
Satellite measurements come from two sources; sea surface temperature and salinity are
provided by the Physical Oceanography Distributed Active Archive Centre (podaac.jpl.nasa.gov),
and surface height anomalies are provided by the Copernicus Marine Environment Monitor-
ing Service (marine.copernicus.eu).
In Situ measurements used are available from 3 sources; Argo measurements through
Global Ocean Data Assimilation Experiment (usgodae.org), SeaGliders through the School of
Ocean and Earth Science and Technology at the University of Hawai'i at Mānoa (hahana.soest.hawaii.edu/seagliders),
and CTDs through the Hawai'i Ocean Time-Series project (hahana.soest.hawaii.edu/hot).
Reanalysis output is produced as 3-hourly snapshots of the 3D fields temperature, salin-
ity and velocities, as well as the 2D sea surface height field for the full time period. This data
are archived through PacIOOS and can be made available for research purposes.

## Acknowledgements

The authors would like to thank the GODAE for hosting the Argo observations and the
HOT project for CTD and SeaGlider data. The authors would also like to thank Y.L. Chen of
the University of Hawai'i Department of Meteorology for the atmospheric model data MM5
and WRF. The authors are grateful to two anonymous reviewers and the editor for helping im-
prove this paper.

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

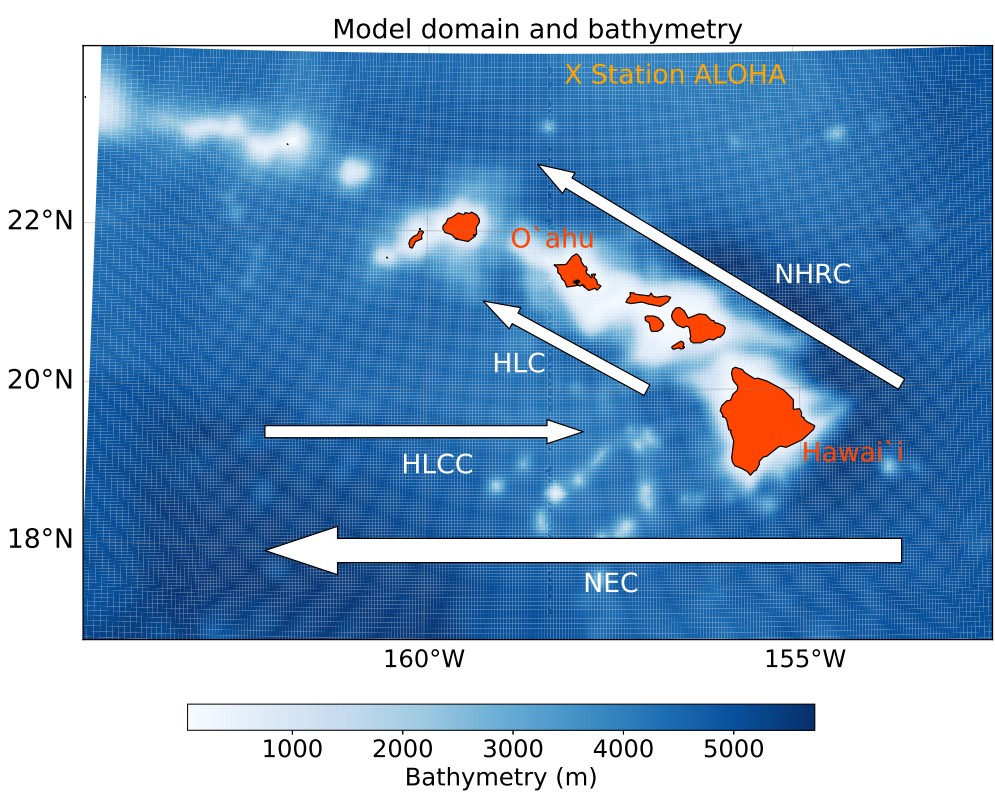

 **Figure 1.** Model domain and bathymetry, with mean currents labelled from *Lumpkin and Flament* [2013].

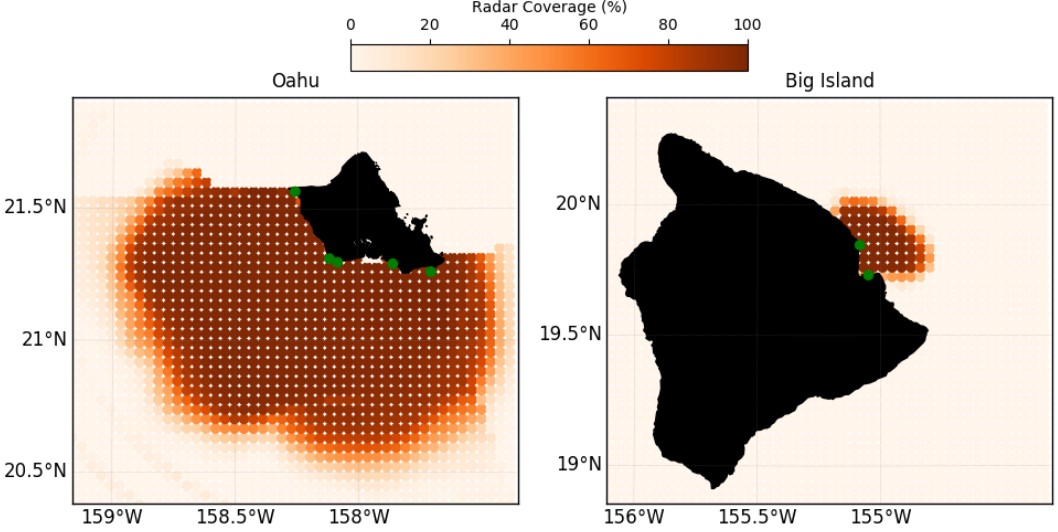

**Figure 2.** Composite image of percentage coverage for all radar sites (situated at green dots) when all are

operational. Where two sites overlap the greater value is taken to indicate the level of coverage at each point.

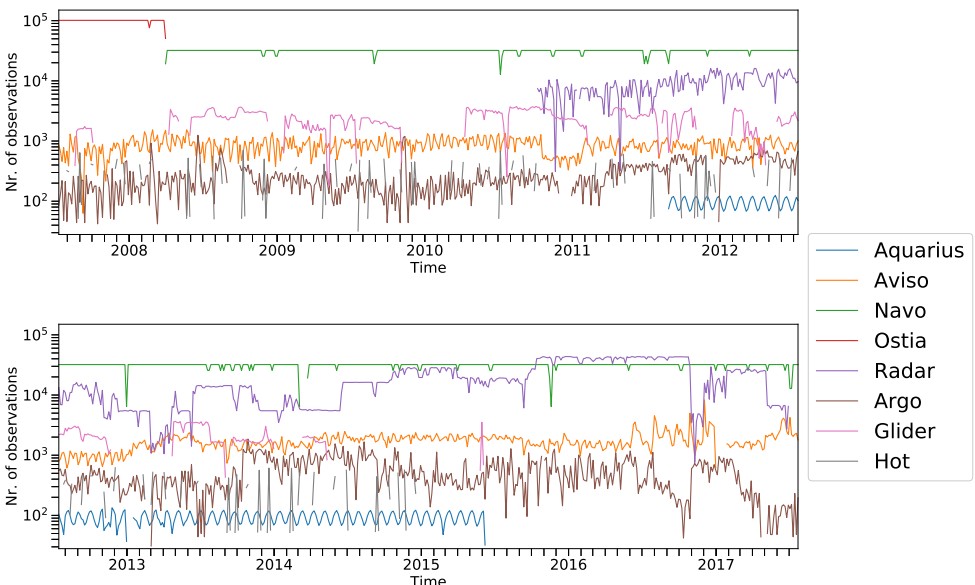

**Figure 3.** Number of observations used within data assimilation run. Note that there tend to be orders of magnitude more satellite or remotely-sensed observations than *in situ*.

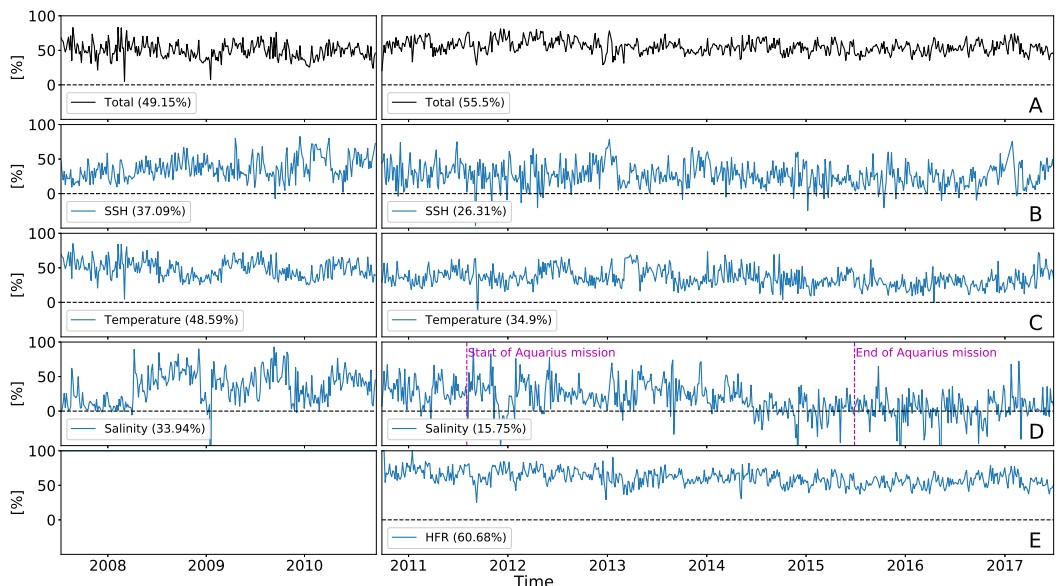

**Figure 4.** Time-Series of percentage reduction in the I4D-Var cost function; Left column are pre-HFR observations, right post-HFR, with the mean value given in parentheses. Dashed lines mark the limit of 0, below which there is no reduction in the cost function for that variable. A) Total cost function reduction for all observations; B) Sea surface height observations, C) Temperature observations; D) Salinity observations; E) HFR observations.

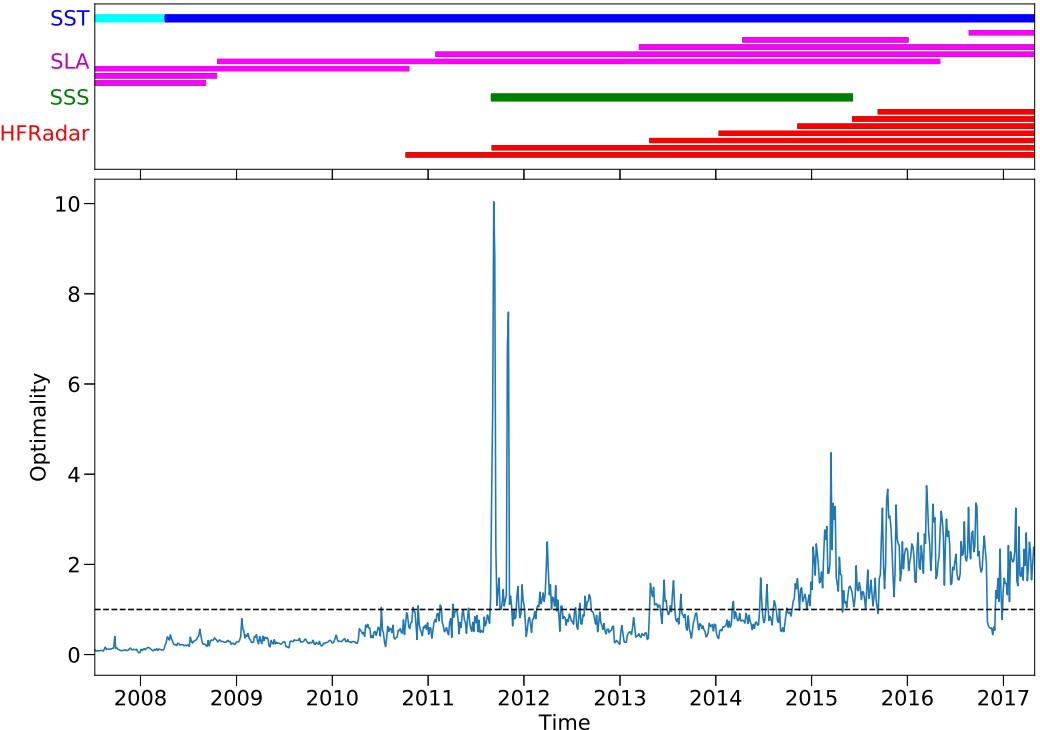

 **Figure 5.** Top - Gantt chart of remotely sensed observations used in the study. Bottom - Optimality of

 I4D-Var data assimilation with the dashed line representing the theoretical minimum.

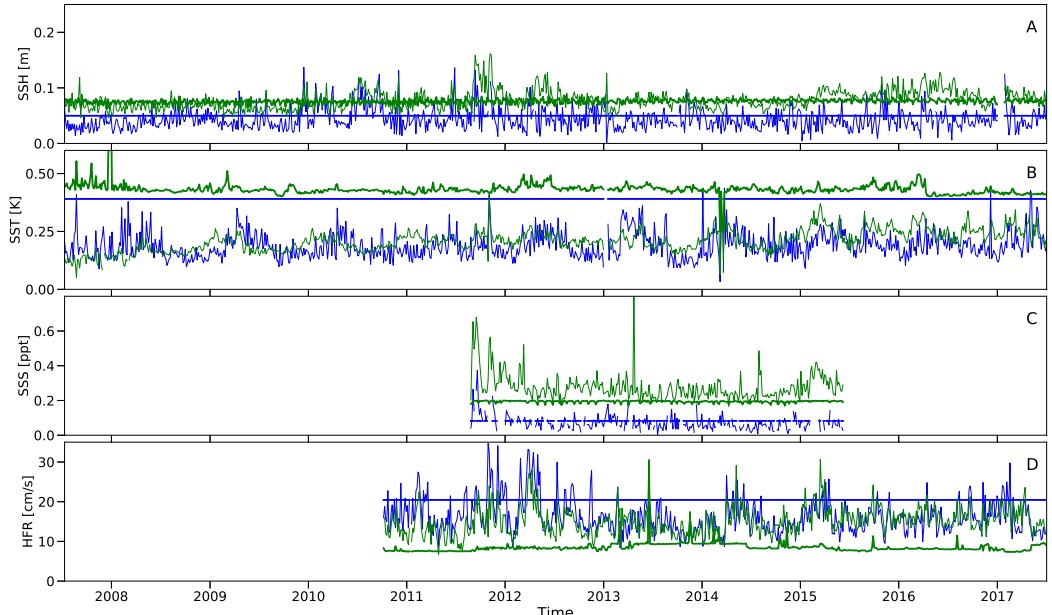

<sub>770</sub> **Figure 6.** Time series of spatially averaged background (blue) and observation (green) errors, with thick

<sub>771</sub> lines showing *a priori* values and thin lines the posterior calculated using Equations (5) and (7). A) Sea

<sub>772</sub> Surface Height; B) Sea Surface Temperature; C) Sea Surface Salinity and D) HFR.

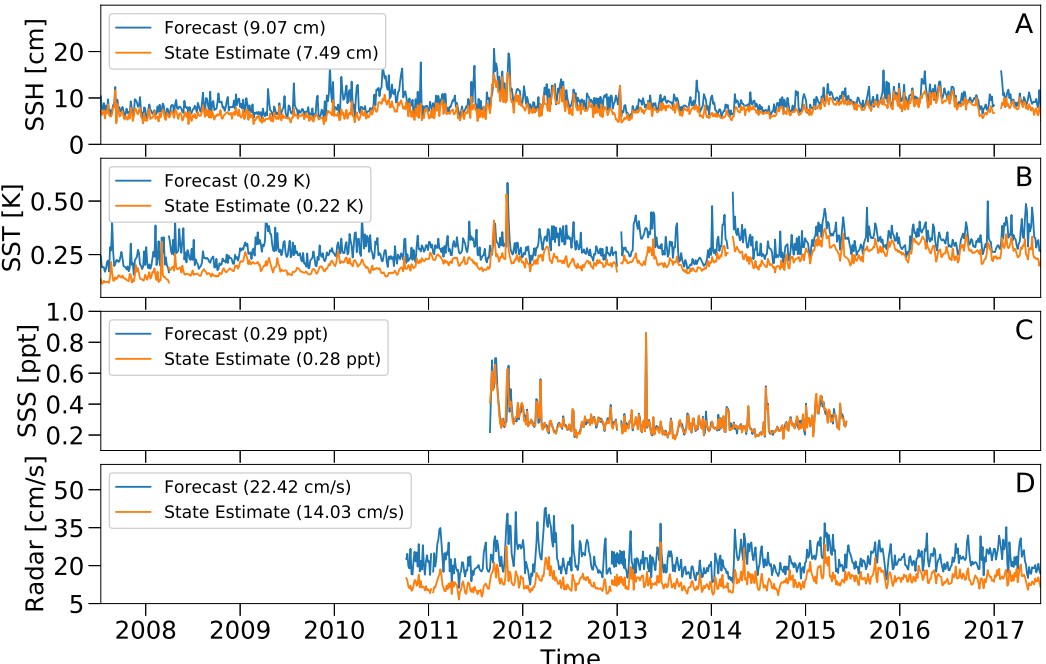

**Figure 7.** Time series of root mean squared anomalies (RMSA) between remotely sensed observations and two model realizations; the state estimate (orange) and the forecast (blue). A) Sea Surface Height; B) Sea Surface Temperature; c) Sea Surface Salinity and D) HFRs

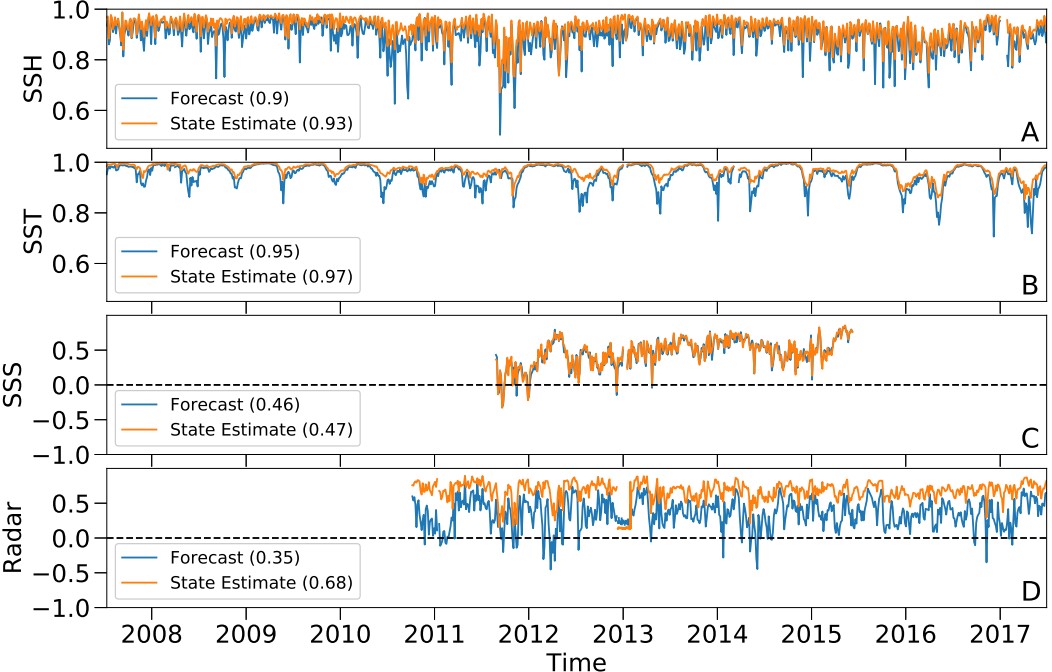

**Figure 8.** Time series of anomaly correlation coefficients (ACC) between remotely sensed observations

and two model realizations; the state estimate (orange) and the forecast (blue). A) Sea Surface Height; B) Sea

Surface Temperature; c) Sea Surface Salinity and D) HFRs

.

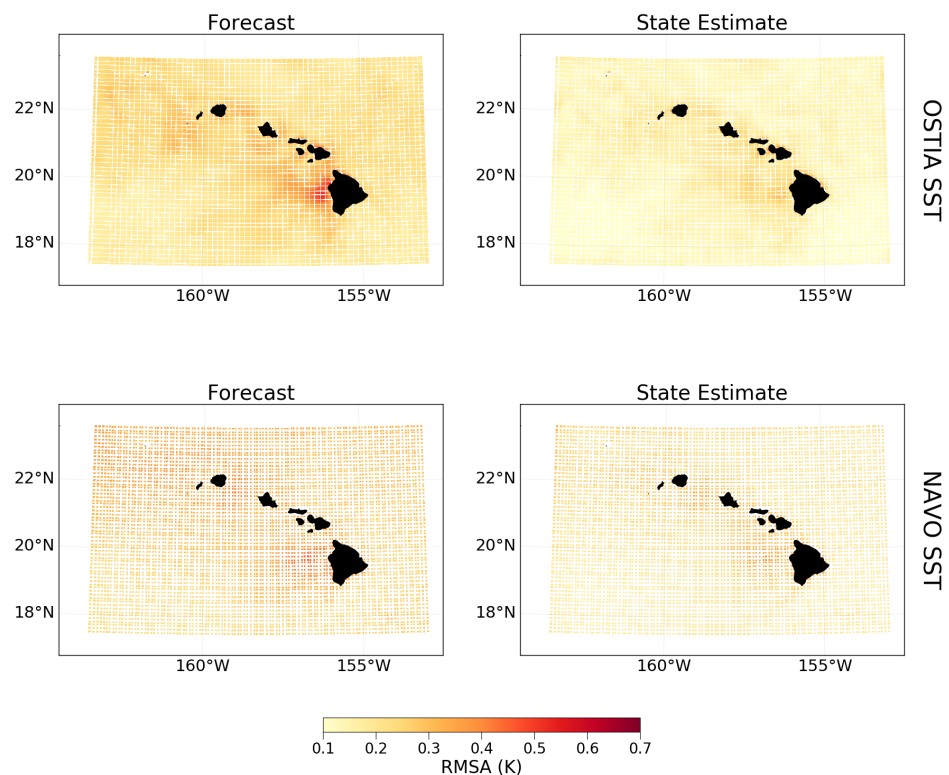

**Figure 9.**    Spatial maps of RMSA for SST observation sources for the forecast (left) and the state estimate (right). Top - OSTIA data (2007-2008); Bottom - NAVO data (2008-2017). The typical error of representativeness is around 0.4 K.

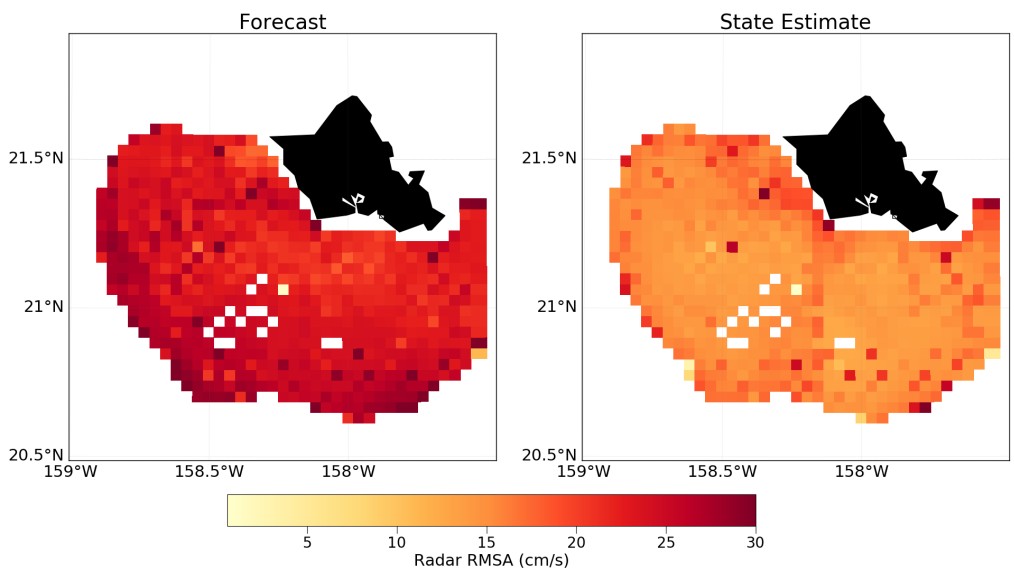

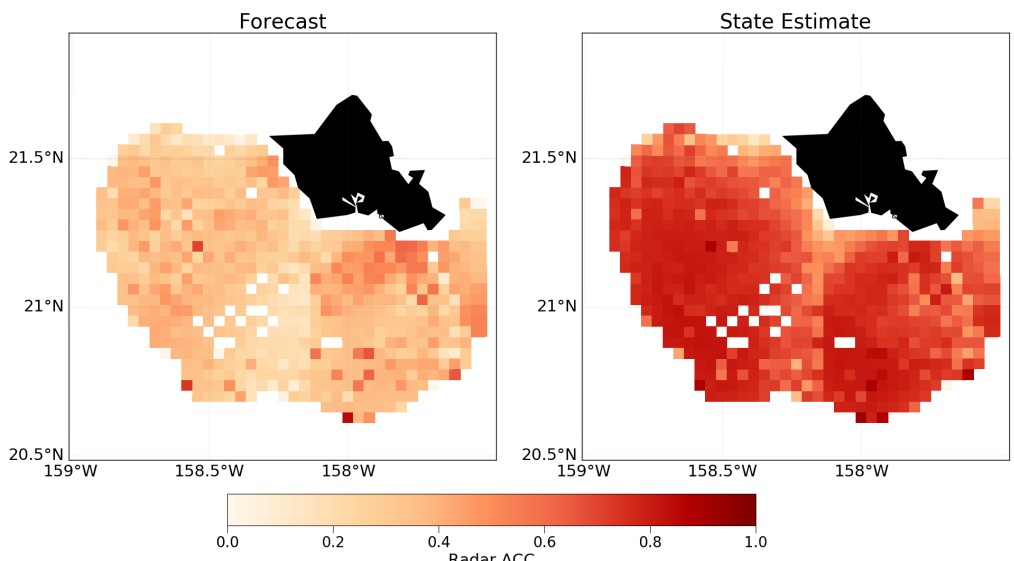

 **Figure 10.** Spatial maps of HFR statistics for south Oʻahu for the forecast (left) and the state estimate

 (right). Top panel: RMSA; bottom panel: ACC.

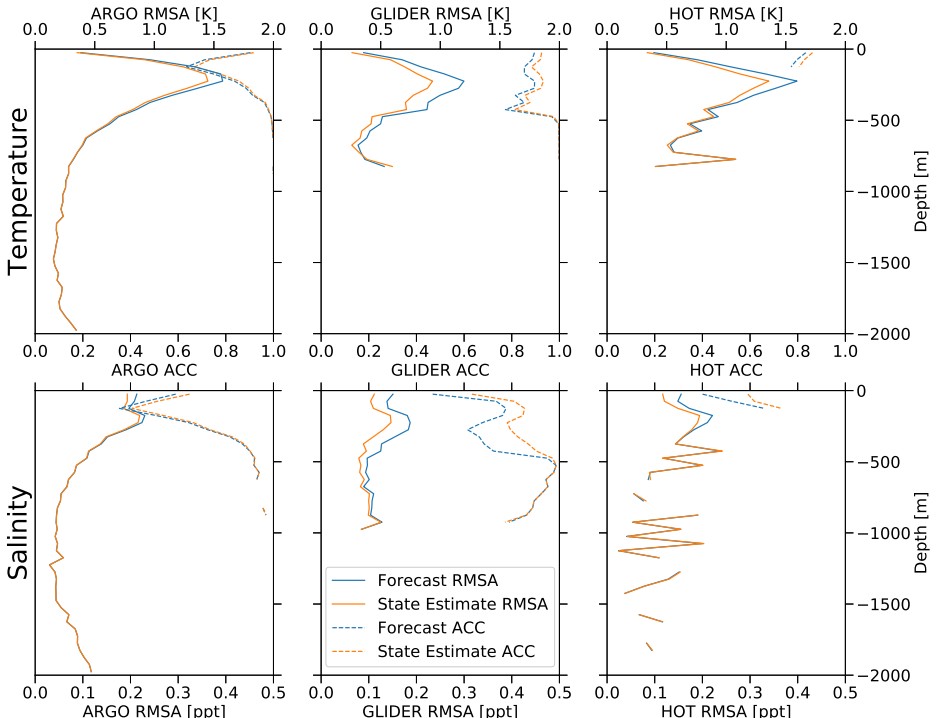

**Figure 11.** RMSA (solid) and ACC (dashed) profiles of subsurface temperature (top) and salinity (bottom)

for Argo floats, SeaGliders and HOT CTDs for the forecast (blue) and the state estimate (orange). Data were

binned into $50\ m$ intervals.

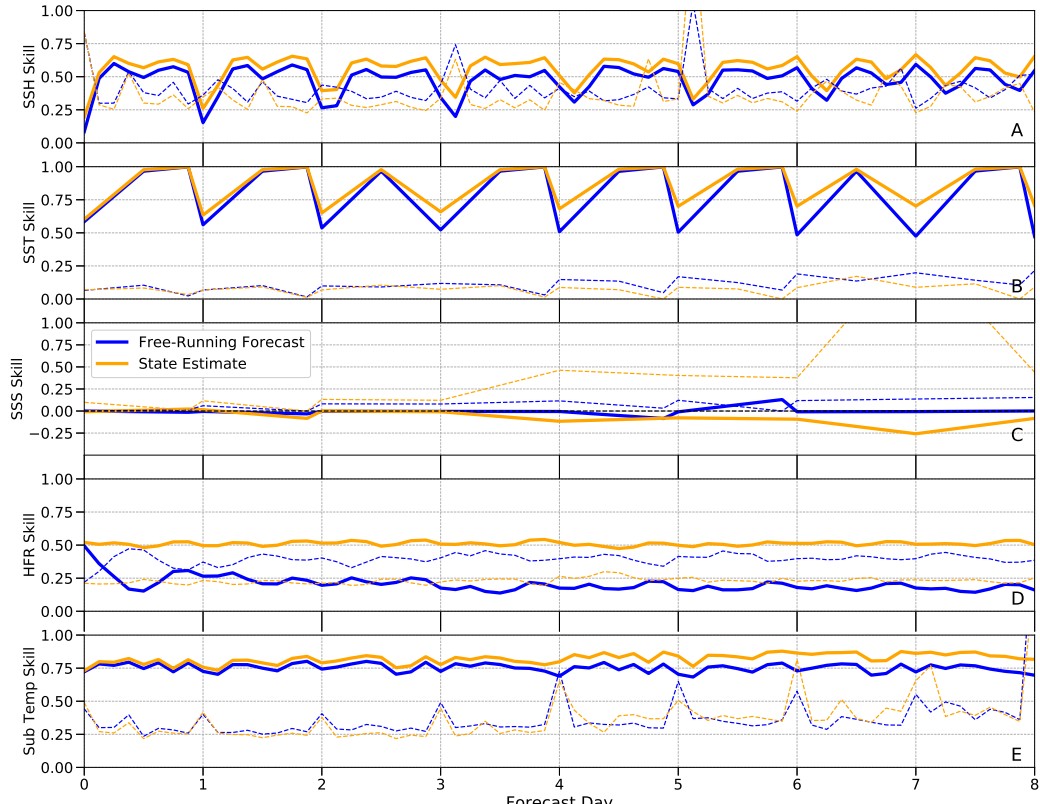

**Figure 12.** Mean skill metric for remotely sensed observations as a function of forecast length. Solid lines:

Skill (see equations 10 and 11); dashed lines: standard deviation of skill. A) Sea Surface Height; B) Sea

Surface Temperature; C) Sea Surface Salinity; D) HFRs and E) subsurface temperature

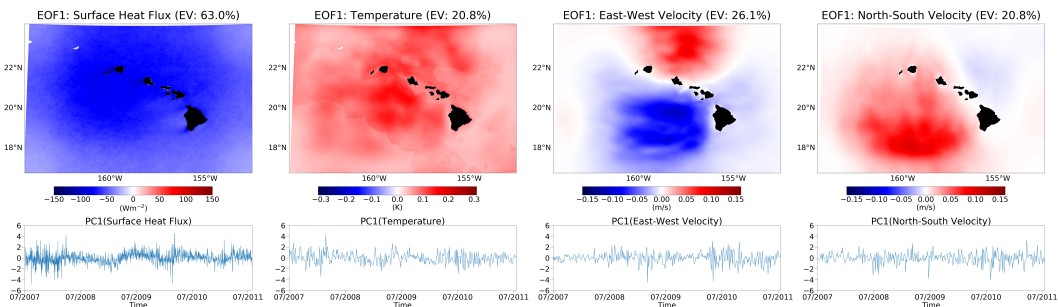

 **Figure 13.** EOF1 and PC1 of initial condition increments for temperature, east-west velocity and north-

 south velocity (all averaged 0-100 m) and of forcing perturbations applied to surface heat flux.

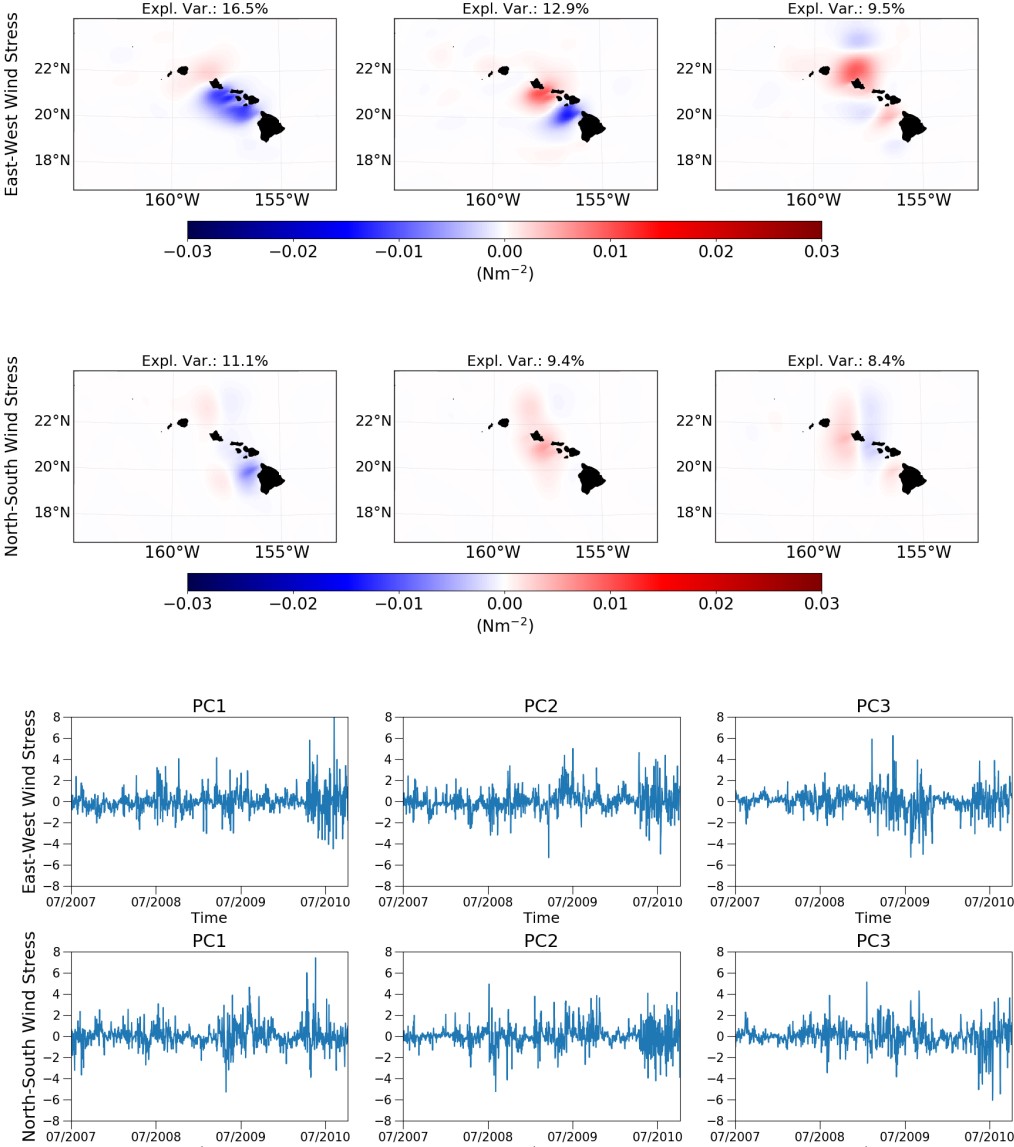

**Figure 14.** Spatial EOF patterns and principal components (PC) of wind stress perturbations for the period prior to the assimilation of HFR measurements (June 2007 - September 2010). The EOFs were calculated using the routines described in *Dawson* [2016].

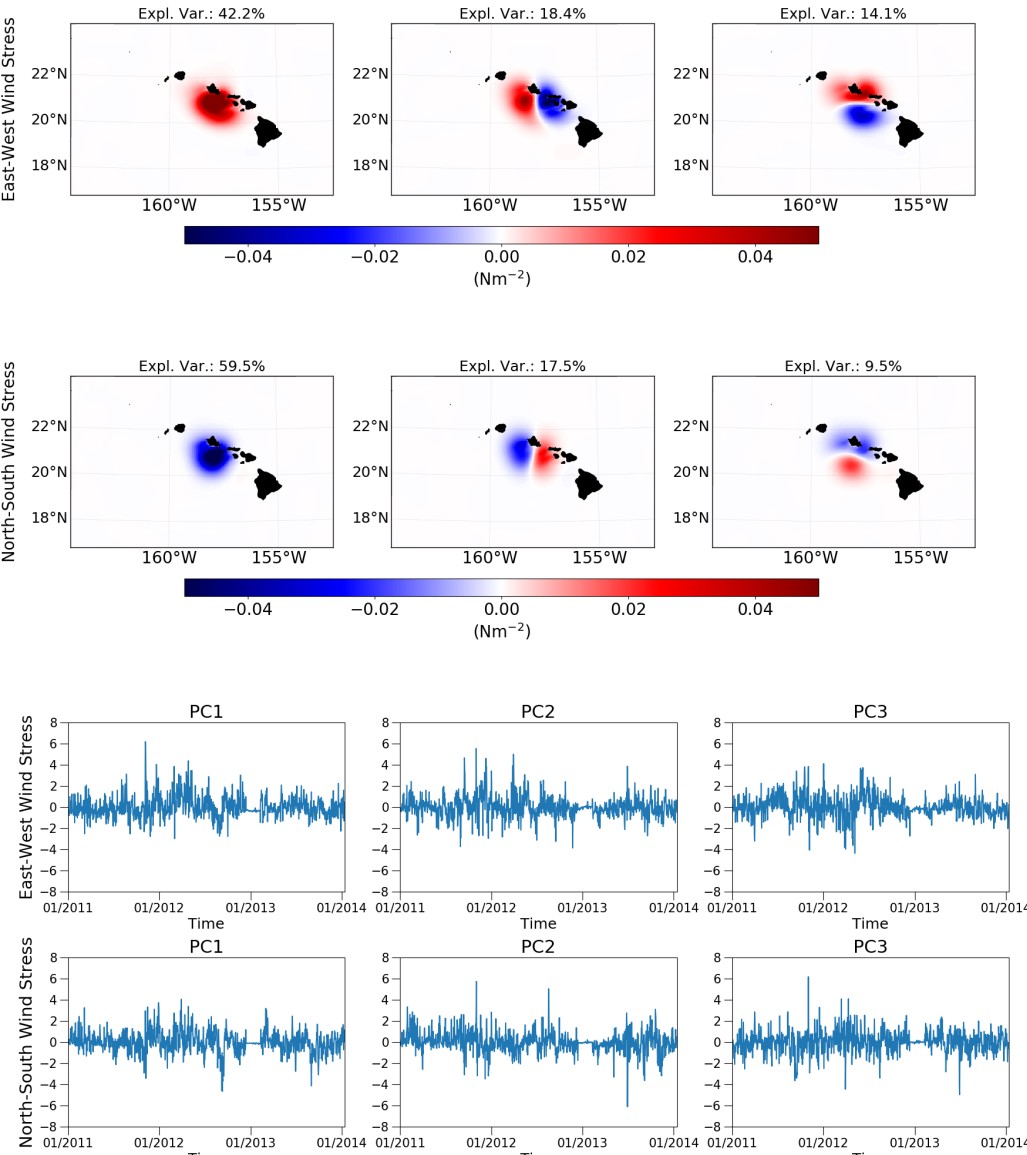

**Figure 15.** Spatial EOF patterns and principal components (PC) of wind stress perturbations for the period including the assimilation of HFR measurements (January 2011 - January 2014).