# Peer review of "Reanalysis of the PacIOOS Hawaiian Island Ocean Forecast System, an implementation of the Regional Ocean Modeling System v3.6"

_Geoscientific Model Development, 2018_

## Short Comment (SC1) · 9 Jul 2018

Dear authors,

referring to the Code and Data Availability section: please be more precise about how to access the exact version of the model used for your publication.

As explained in https://www.geoscientific-model-development.net/about/manuscript_types.html. GMD is encouraging authors to upload the program code of models (including relevant data

sets) as supplement or make the code and data of the exact model version described in the paper accessible through a DOI (digital object identifier). In case your institution does not provide the possibility to make electronic data accessible through a DOI you may consider other providers (eg. zenodo.org of CERN) to create a DOI. Please note that in the code availability section you can still point the reader to how to obtain the newest version. If for some reason the code and/or data cannot be made available in this form (e.g. only via e-mail contact) the "Code Availability" section need to clearly state the reasons for why access is restricted (e.g. licensing reasons).

Especially, please note, that it is not enough, that the code will be available in the future. It must be available now and the exact version of the code published in this article needs to be made available.

Best regards, Astrid Kerkweg
* * *

---

## Referee Comment (RC1) · Anonymous Referee #1 · 26 Jul 2018

The paper analyses an implementation of the Incremental Strong Constraint Variational Data Assimilation scheme within the ROMS model system to generate a reanalysis for the Hawaiian Archipelago. In particular, the improvements in the system due to a newer version of the model source code and modifications in the data processing in comparison to a free running simulation are evaluated. However, the modifications to the model source that can be relevant to the results are not discussed and the modifications to the observations processing not clear. While the analysis techniques are appropriate, they are not novel. Although the work is interesting and provides information that can

be useful for researchers developing similar systems for other regions, there are analysis flaws and misconceptions that must be addressed before it is ready for publication. The reanalysis evaluation is made against a free running simulation, and based on the same set of observations assimilated in the reanalysis. Therefore the results should be considered as an evaluation of the assimilation scheme, and not as improvements to the system. A comparison to previous versions of the reanalysis and/or global high resolution products (such as the GLORYS reanalysis) is necessary to evidence any ocean estate representation gain by the present simulation. This is useful to justify the effort of implementing and maintaining a regional operational system. Moreover, there is a lack of physical interpretation of the simulation misfits and the increments to the initial and boundary conditions. I recommend the authors are given the opportunity to improve their manuscript through a MAJOR REVISION before it is accepted for publication in the Geophysical Model Development Journal.

MAIN COMMENTS:

While the paper is partly based on the upgrade to a newer version of ROMS, the latest version is not used. The study uses ROMS 3.6 while the version 3.7 was released over 4 years ago, and the current version is 3.8. The authors must justify why an older version of the model code is used.

Since an improvement is claimed for the current reanalysis, there should be a comparison against the previous version of the Hawai'i reanalysis in order to identify where and why such gain occurs.

Although the comparison against a free running model that constitutes a downscaling of a global reanalysis is interesting, the manuscript would benefit from a comparison against the global reanalysis itself. This can provide an estimation of the improvement in relation to the solution used to force the open boundaries, and justify the effort of implementing a regional assimilation system.

How are the super-observations generated? Is there a projection/average over the

model grid? This is especially important for the HFR since it presents higher resolution than the model – and may therefore contain processes unresolved by the simulation.

The case for salinity is curious in Figure 4. There are "negative improvements" even during the period covered by Aquarius mission. Figures 7 and 8 reinforce the idea that there is no benefit in assimilating SSS. Moreover, I wonder why other SSS observations were not explored after the end of the Aquarius mission (SMOS). Since the assimilation of SSS is not a common feature in ocean reanalysis and operational systems, a more detailed analysis of the reasons behind this relatively poor performance would improve the interest on the manuscript.

In sections 3.2 and 3.3 you show the a priori SST errors are overestimated for both the observations and the background model solution. Please elaborate on how this impacts your reanalysis results. Did you make any experiments changing the errors estimation method? The same applies to the HFR (underestimated errors).

Although the comparison against the assimilated observations is interesting, it basically diagnoses the assimilation is working. However, it says nothing about any improvement in the representation of the regional dynamics. For this purpose, the model must be compared against independent observations (not assimilated). In special, observations in under-sampled areas/variables would be interesting. Perturbations generated during the estate estimate process can degrade the solution in regions where no constraint is provided due to the lack of observations. Are there any other observations that can be used for such (WOD, drifters, SMOS, etc)? If no independent observations can be found, the authors should clearly state the limitations of the provided comparisons.

It is interesting to note there is a good agreement between the variability of both RMSA and CCA from the forecast and the analysis for all observations in Figures 7 and 8 (especially the low frequency). This suggests a physical process may be absent from the background solution, represented by the forecast. Although the assimilation reduces the mismatch, it does not introduce the missing physics. Please comment on this.

In the evaluation of the analysis of increments in the initial conditions, I don't understand why you averaged over the top 100m. By one side most of the data is in the surface (SST), by the other side there is great interest in what happens below the mixed layer. Especially after you show the larger mismatches are at the thermocline. For both sections 6.1 and 6.2, a deeper interpretation of the physical meaning of the EOFs is lacking. This kind of analysis can give you an insight into processes that are not well represented by the background solution, and fuel improvements in the system. The inclusion of the temporal components of the EOFs should help such interpretation.

DETAILED COMMENTS:

Lines 106 – 113: Please comment on the compatibility between the 2 sources for atmospheric forcing used. Are there discontinuities in the fluxes from the different sources? How does the ocean model respond to this change?

Line 122: Please explain in more details how the observations errors (uncertainty) are calculated and the assumptions made. There are several possible methods for doing so, and the outcomes can greatly impact the assimilation results. (you already started this in lines 194-196)

Line 129: Were sensibility tests conducted to estimate the optimum number of inner/outer loops so J is properly minimized? Can the authors present a plot (probably as a supplementary figure) to justify their choice?

Lines 140-141: What is considered as "close to the boundary" and "shallow water"? The values should be given.

Lines 143-160: Please justify the change in the SST data source to a coarser product in April 2008.

Line 205: What do you mean by that the HFR data is less reliable at the range limit. Does it mean there was a different treatment for such data? Or are you referring to a smaller number of observations?

Lines 231-233: You should briefly explain how the breakdown of the cost-function is made, and offer references for further details.

Line 232: Probably you meant model variable by "observation type".

Line 233: change "is" for "it"

Lines 259-261: It is not clear what the "optimality value" is. If it is the result of equation 1, would this mean that a optimality of 1 refers to the model perfectly representing the observations? Please be more straightforward about it.

Paragraph bellow line 307 (has no line numbers): Are the initial conditions for the analysis and the forecast the same? You specifically say that "the boundary and atmospheric forcing are altered as part of the state estimate solution". However, the modification of the initial conditions is usually the most important factor impacting the model solution. Please clarify what are the initial conditions used by the forecast (updated or otherwise).

Line 393: The explanation of the skill metrics is confusing. What is the persistence assumption? Please explain it better. Moreover, if the assimilation window is 4 days how do you calculate the analysis skill for 8 days?

Lines 410-412: The fact there is a reduction in skill once per day for the SST is highly dependent on how you define the persistence. What would happen if it was defined from the mean of the first day? Would it be a more balanced option? Please comment.

Lines 420-422: Could you please provide some evidence the perturbations of the boundary conditions don't influence the model solution? Or at least references of similar systems that provide such evidence.

Lines 451-452: How was the update interval for the perturbations in the forcing chosen? What was the reasoning behind the 6 hours value?

Lines 470-472: Was a flux correction (QCORR) applied to ROMS?

Line 475: Please substitute "vertically" and "horizontally" by "meridionally" and "zonally".

Section 5: It is surprising how the analysis represents a small improvement in performance when compared to the forecast. Also, it seems the forecast skill is very stable along the 8 days window. Please elaborate on the reasons behind this.

Sections 6.1 and 6.2: Please include the temporal (different cycles) component of the EOFs. These are indispensable for the interpretation of the results.

Figure 3: Do you know the reason for the "sinusoidal" variation on the number of Aquarius observations?

Figure 11: Interesting how the assimilation can't reduce the RMSA or improve the CCA bellow 500m for the Argo and HOT data. But the same doesn't happen for the gliders data. Can you articulate about why this happens?

Figure 12: The standard deviation around the mean SS should be presented. Also, a grid would make the plot easier to read.

Figures 13 and 14: Please include the time component of the EOFs.

---

## Referee Comment (RC2) · Anonymous Referee #2 · 14 Aug 2018

General comments:

This paper presents a 10-year reanalysis of the Hawaiian Islands region, using IS4DVar and assimilating a variety of observations. The authors show that the state-estimates provide an improved representation of the (assimilated) observations compared to the forecasts, which is to be expected. It is unfortunate that they do not present comparison to any independent observations.

The authors mention in the introduction and the summary that the system presented

uses an updated model and data assimilation scheme, but do not elaborate as to how the model has been updated nor do they quantify the improvements. Mention of this 'updated version' should be removed unless they are going to provide some quantitative comparison along with a description of the updates.

The authors present EOFs of the increment adjustment to the initial conditions and surface forcing, however physical interpretation of the increments is lacking.

In general, the model and data assimilation methodology presented is sound, but the manuscript lacks insight that the authors can gain from their work. How do the results help to understand and improve their model and assimilation system?

I recommend that the manuscript be returned to the authors for MINOR revision before acceptance for publication in the Geoscientific Model Development Journal.

Specific comments:

Line 13: remove the first HFR

Line 52: The HLCC is perhaps a weak current, but it is not a small current (in its spatial extent).

Figure 1: I wonder if it would be useful to label the islands, as you refer to them quite a bit in the text.

Lines 127-128: The inner loops are performed before the NL model is updated in the outer loop.

Lines 140-141: How close and how shallow?

Section 2.3: The author does not explain if super-observations are used and how the observation error variances are specified.

Line 201: Call the Big Island Hawai'I instead for consistency throughout the paper.

Line 206: . . . around Hilo Bay (on the northeast of the island) . . .

Line 207: And what happens if it's <80%, is it still used?

Line 208 on: Do you grid in space to get super-observations?

Figure 3: Hard to read the legend

Figure 4, 6: Legends and text are too small to read

Line 246: and have higher errors . . .?

Line 243-245: This statement is very vague.

Line 305: . . . better forecasts than other methods that perturb the state at single times. These methods may better reduce. . ... , but can add . . . . . .

Section 4: Comparisons are only made against observations that are assimilated. This limitation needs to be made clear in the section introduction. Are there not any other observations of the region that could provide comparison to independent observations (e.g. ship-based CTD observations) ?

Figure 9 and line 344: Do you mean the atmospheric model has inaccuracies in the representation of the heat fluxes? Or something else? Is this evident in the surface forcing increments discussed later?

Line 356 on: Did you look at how these adjustments extended beyond the radar coverage regions? Some snapshot examples might be nice to elucidate how the currents deviated in the forecast and how they were improved in the state-estimation (as well as how it looked beyond the coverage region). This may help add some physical context to this analysis.

Line 371-372: This sentence is unclear.

Line 380: It's not much worse, I would say "of the same magnitude".

Line 385-387: Is this because the model background errors are low and the observation errors are relatively high below 500m, so the state-estimate makes little adjustment to

salinity below 500m?

Figure 12D: Interesting that the forecast skill degrades within the first 12 hours for the radials.

Lines 431-433. By 'background values' do you mean the background standard deviations? It is surprising that the SSH is adjusted so little. Does the relatively large adjustments to the velocities suggest the system may be over fitting to the HF radar observations? Can you comment on the relative increments for the different variables as a percentage of the backgound standard deviations?

Line 435: Positive everywhere, so this suggests a bias that is being corrected. Can you comment on bias?

Line 437: Do you mean horizontal temperature gradients? Can you explain how this dynamical characteristic relates to the higher increments in this region.

Line 446: increasing or decreasing (you don't show the temporal expansion function which could have both negative and positive sign)

Figure 14 a) mode 1 is all negative. Again, can you talk about bias.

Lines 491-493: it is not clear if the improvement you are referring to is relative to the 'older' model version, or relative to the forecasts (which is the comparison that has been made throughout the paper).

———————————————————

---

## Author Comment (AC2) · 7 Oct 2018

**Response to Reviewer #1**

*We thank the reviewer for the thoughtful and thorough review of the manuscript. We have made a major revision to the manuscript to address all of the reviewer's comments, and we believe that we have addressed the concerns raised. We accept all of*

*the reviewer's comments.*

The paper analyses an implementation of the Incremental Strong Constraint Variational Data Assimilation scheme within the ROMS model system to generate a reanalysis for the Hawaiian Archipelago. In particular, the improvements in the system due to a newer version of the model source code and modifications in the data processing in comparison to a free running simulation are evaluated. However, the modifications to the model source that can be relevant to the results are not discussed and the modifications to the observations processing not clear. While the analysis techniques are appropriate, they are not novel. Although the work is interesting and provides information that can be useful for researchers developing similar systems for other regions, there are analysis flaws and misconceptions that must be addressed before it is ready for publication. The reanalysis evaluation is made against a free running simulation, and based on the same set of observations assimilated in the reanalysis. Therefore the results should be considered as an evaluation of the assimilation scheme, and not as improvements to the system. A comparison to previous versions of the reanalysis and/or global high resolution products (such as the GLORYS reanalysis) is necessary to evidence any ocean estate representation gain by the present simulation. This is useful to justify the effort of implementing and maintaining a regional operational system. Moreover, there is a lack of physical interpretation of the simulation misfits and the increments to the initial and boundary conditions. I recommend the authors are given the opportunity to improve their manuscript through a MAJOR REVISION before it is accepted for publication in the Geophysical Model Development Journal.

MAIN COMMENTS: While the paper is partly based on the upgrade to a newer version of ROMS, the latest version is not used. The study uses ROMS 3.6 while the version 3.7 was released over 4 years ago, and the current version is 3.8. The authors must justify why an older version of the model code is used.

*Co-Author Powell is a ROMS code contributor, and the ROMS 3.6 codebase was used with modifications for the HF radar assimilation and others mentioned in Souza et al. (2015). The most current "stable" release of ROMS is the 3.6 branch as listed in the official release announcements at*
https://www.myroms.org/forum/viewforum.php?f=13.
*The current development release is considered 3.7, and it includes the HF code developed by Co-Author Powell. The primary differences between ROMS 3.7 and ROMS 3.6 are the inclusion of the online nesting routines that are not utilized in this paper.*

Since an improvement is claimed for the current reanalysis, there should be a comparison against the previous version of the Hawai'i reanalysis in order to identify where and why such gain occurs.

*The original text was rather vague in reference to "improvement", rather the suggestion is that we are conducting a decadal reanalysis of the Hawaiian region using the configuration as detailed in citesouza2015. The test period used in Souza et al. (2015) was only an 18 month test case. This paper uses all available data over 10 years. The manuscript has been changed accordingly*

Although the comparison against a free running model that constitutes a downscaling of a global reanalysis is interesting, the manuscript would benefit from a comparison against the global reanalysis itself. This can provide an estimation of the improvement in relation to the solution used to force the open boundaries, and justify the effort of implementing a regional assimilation system.

*The authors tried to gain access to the GLORYS data mentioned by the reviewer. The data, however, are not public and access comes with cost. Furthermore, the global systems use statistical, time-invariant assimilation schemes in an attempt to replicate observations. These schemes work to minimize the residual between the model and observations, but accomplish this at a cost of non-dynamical increments that typically worsen forecast skill. We have revised the manuscript to discuss that in 4D-Var, the purpose is to* represent *all available data in time and space not* replicate*. The only reason to withhold data is because the statistics do not fully account for the time variability. With state estimation techniques (4D-Var and ENKF), the statistics are time evolving by the dynamical system and all observations are relevant and used.*

How are the super-observations generated? Is there a projection/average over the model grid? This is especially important for the HFR since it presents higher resolution than the model – and may therefore contain processes unresolved by the simulation.

*For most datasets, we do not use "super-observations". In a least-squares minimization, finding the minimum variance of co-located values is the same as averaging them and assigning a larger variance. That said, for HFR, we did combine some due to the large number of observations; however, oftentimes, we have only one in a single grid cell. Regions where there are more than one HF radial in a single grid cell are those nearer to the HFR source. As such, the angles are very close, and we tested averaging the angle and radial value versus projecting them. Some of the details are found in Souza et al. (2015).*

The case for salinity is curious in Figure 4. There are "negative improvements" even during the period covered by Aquarius mission. Figures 7 and 8 reinforce the idea that there is no benefit in assimilating SSS. Moreover, I wonder why other SSS observations were not explored after the end of the Aquarius mission (SMOS). Since the assimilation of SSS is not a common feature in ocean reanalysis and operational systems, a more detailed analysis of the reasons behind this relatively poor performance would improve the interest on the manuscript.

*We thank the reviewer for calling this out. Due to the loss of two SeaGliders in 2014, the SeaGlider data (temperature and salinity) stopped in mid-2014. After this period, only sporadic Argo as well as the SSS fields were available. The SSS fields are highly noisy (0.2 ppt) with coarse spatial coverage (less than 100 observations used for our region). As such, there were very few salt measurements available after mid-2014. These have little impact on the cost-function to minimize; hence, there are—in a practical sense—no constraints for salt. We have updated the manuscript accordingly.*

In sections 3.2 and 3.3 you show the a priori SST errors are overestimated for both the observations and the background model solution. Please elaborate on how this impacts your reanalysis results. Did you make any experiments changing the errors estimation method? The same applies to the HFR (underestimated errors).

*It is true that the errors were overestimated, which means that there is an "underfitting" to the data (that is, it is less trusted); however, this is somewhat intentional as neither SST or HFR observations are linearly independent. The purpose is to assimilate the observations with the errors that are ascribed to them. In the case of the SST, if the data source attaches a 0.4K error to the value, but the posterior analysis suggests that is higher than it should be, it is because of the lack of linear independence. As shown in Matthews et al. (2011), SST observations within approximately 100km in this region are correlated. It is true that changing the errors would provide a different analysis; however, one has to be careful that the point is not to tune or optimize the posterior error estimates, but to optimize the representation of the observations as a whole in space and time. More can be found in a number of papers by Desroziers et al. (2001, 2005, 2009). We have update the manuscript to point out the work by Desroziers.*

Although the comparison against the assimilated observations is interesting, it basically diagnoses the assimilation is working. However, it says nothing about any improvement in the representation of the regional dynamics. For this purpose, the model

must be compared against independent observations (not assimilated). In special, observations in under-sampled areas/variables would be interesting. Perturbations generated during the estate estimate process can degrade the solution in regions where no constraint is provided due to the lack of observations. Are there any other observations that can be used for such (WOD, drifters, SMOS, etc)? If no independent observations can be found, the authors should clearly state the limitations of the provided comparisons.

*In general, we agree with the reviewer. However, for a robust validation a significant number of independent observations would be required. A handful of e.g. CTD profiles would not be reliable. All data available to the authors have been used for the assimilation. The authors refrained from reserving data sets for validation as this is an arbitrary procedure. This limitation is now clearly mentioned in the revised manuscript in Section 4 and in the Summary.*

It is interesting to note there is a good agreement between the variability of both RMSA and CCA from the forecast and the analysis for all observations in Figures 7 and 8 (especially the low frequency). This suggests a physical process may be absent from the background solution, represented by the forecast. Although the assimilation reduces the mismatch, it does not introduce the missing physics. Please comment on this.

*Please recall that the assimilation is 4 days and the figures span 10 years. The "variability" is the variations in the number of observations, locations, etc. that are different between each window. The lower frequency in SST, particularly, the twice yearly increase in RMSA and drop in CCA (relatively low amplitudes), are due to the warming/cooling of the seasons, and disconnect between the HYCOM boundary conditions and the SST used. There are some physical processes that are not resolved that commonly are addressed that can be seen in Figure 9. In the lee of the Big Island, the lack of winds (physically resolved) and cloud cover (not physically resolved) drives a strong diurnal cycle in the SST. This is captured in the observations, but the modeled diurnal amplitude is smaller.*

In the evaluation of the analysis of increments in the initial conditions, I don't understand why you averaged over the top 100m. By one side most of the data is in the surface (SST), by the other side there is great interest in what happens below the mixed layer. Especially after you show the larger mismatches are at the thermocline. For both sections 6.1 and 6.2, a deeper interpretation of the physical meaning of the EOFs is lacking. This kind of analysis can give you an insight into processes that are not well represented by the background solution, and fuel improvements in the system. The inclusion of the temporal components of the EOFs should help such interpretation.

*Mixed Layer Depth estimates in the region (both from the model and observations, see Matthews et al. (2012) vary between 120 and 80m throughout the seasons. As such, we settled for a 100 m average. We have revised the discussion of the EOFs. However, it must be kept in mind that by computing the EOFs of the increments and adjustments,*

*we are dealing with EOFs of residuals. These residuals have non-physical variance. In the revised manuscript we discuss the spatial patterns of EOFs but clearly state that no physical modes came out of our EOF analysis.*

DETAILED COMMENTS: Lines 106 – 113: Please comment on the compatibility between the 2 sources for atmospheric forcing used. Are there discontinuities in the fluxes from the different sources? How does the ocean model respond to this change?

*We have revised the manuscript to better explain the generation of the pre-WRF atmospheric dataset. They are both made to be consistent with the NCEP/NCAR reanalysis; hence, though the local variability may be different the large-scale structure and fluxes are the same.*

Line 122: Please explain in more details how the observations errors (uncertainty) are calculated and the assumptions made. There are several possible methods for doing so, and the outcomes can greatly impact the assimilation results. (you already started this in lines 194-196)

*As noted above, we do not use "super-observations" for most of the observational*

*types (because the resolution of the model is similar or greater than many of the observational platforms. For the* in situ *observations, they are combined in the vertical every 10 m. The prescribed errors are described in the manuscript, as the error of representativeness is calculated using long-term analyses of the vertical profiles.*

Line 129: Were sensibility tests conducted to estimate the optimum number of inner/outer loops so J is properly minimized? Can the authors present a plot (probably as a supplementary figure) to justify their choice?

*These were conducted in earlier experiments, see Matthews et al. (2012) and Souza et al. (2015)*

Lines 140-141: What is considered as "close to the boundary" and "shallow water"? The values should be given.

*Due to the sponge layer at the boundary of the domain, observations within one Rossby radius (∼80 km) of the domain's boundary are neglected. This is now specified in the revised manuscript. The word "shallow" has been removed.*

Lines 143-160: Please justify the change in the SST data source to a coarser product in April 2008.

*Though the product is coarser, it is more accurate and preferred. The NAVO SST product is higher quality than the OSTIA interpolated product. The OSTIA product is overly smooth and removes ocean structure. Unfortunately, the NAVO SST was not available prior to 2008.*

Line 205: What do you mean by that the HFR data is less reliable at the range limit. Does it mean there was a different treatment for such data? Or are you referring to a smaller number of observations?

*As the returns are further away from the source, the quality of the return decreases. As such, as described in the manuscript, the errors associated with those radials is much higher (as computed spectrally). There are also a smaller number of observations as the radial spreading of the return lines away from the source.*

[Figure]

Lines 231-233: You should briefly explain how the breakdown of the cost-function is made, and offer references for further details.

*The manuscript has been updated. ROMS reports the contributions to the cost-function as a function of the state variables considered by the I4D-Var scheme. The sum of those terms provides the total cost function. We present them individually to show how each state variable is contributing to the cost (in terms of observational residual and increments). The total cost function is detailed in the references provided in the manuscript.*

Line 232: Probably you meant model variable by "observation type".

*Corrected*

Line 233: change "is" for "it"

*Corrected.*

Lines 259-261: It is not clear what the "optimality value" is. If it is the result of equation 1, would this mean that a optimality of 1 refers to the model perfectly representing the observations? Please be more straightforward about it.

*We improved the explanation of the minimum of the cost function and the associated optimality value. An optimality value of one does refer to a "perfect" representation of the observations for the given number of observations that determine the expected value of the minimum.*

Paragraph bellow line 307 (has no line numbers): Are the initial conditions for the analysis and the forecast the same? You specifically say that "the boundary and atmospheric forcing are altered as part of the state estimate solution". However, the modification of the initial conditions is usually the most important factor impacting the model solution. Please clarify what are the initial conditions used by the forecast (updated or otherwise).

*The line numbers were re-established. The initial conditions come from the end of*

*the previous analysis. They are used for the forecast. Then, increments are made to the initial conditions by the I4D-Var method and the altered initial conditions are subsequently used for the analysis run.*

Line 393: The explanation of the skill metrics is confusing. What is the persistence assumption? Please explain it better. Moreover, if the assimilation window is 4 days how do you calculate the analysis skill for 8 days?

*The manuscript has been updated for clarity.*

Lines 410-412: The fact there is a reduction in skill once per day for the SST is highly dependent on how you define the persistence. What would happen if it was defined from the mean of the first day? Would it be a more balanced option? Please comment.

*The daily reduction is due to the prior mentioned diurnal cycle (which can be  1K in the lee of the island); however, this is the tropics and SST varies little from day to day.*

Lines 420-422: Could you please provide some evidence the perturbations of the boundary conditions don't influence the model solution? Or at least references of similar systems that provide such evidence.

*We have not performed a full dynamical analysis; however, the boundary increments are typically less than one-tenth of the increments made to the initial field. The advective speeds of the NEC and NECC (the two major inflows into the domain) are generously 20 cm/s, such that over the 4 day cycle, the increments would propagate around 70km into the domain. Furthermore, due to the SPONGE layer applied, the increments are diffusive. Similarly, in (Powell, 2017), he performed an observational impact analysis and found that the boundary condition increments have no significance to the solution of the metrics used in the paper.*

Lines 451-452: How was the update interval for the perturbations in the forcing chosen? What was the reasoning behind the 6 hours value?

*Six hours were chosen because the atmospheric forcing from NCEP is on 6 hour increments.*

Lines 470-472: Was a flux correction (QCORR) applied to ROMS?

*Yes, the correction was applied because the WRF model was run with an independent SST estimate, which created a heat flux imbalance between the atmosphere and ocean. Because we are nested within Global HYCOM, the QCORRECTION was calculated based on the WRF 10 and 2m fields with the HYCOM SST.*

Line 475: Please substitute "vertically" and "horizontally" by "meridionally" and "zonally".

*Corrected.*

Section 5: It is surprising how the analysis represents a small improvement in performance when compared to the forecast. Also, it seems the forecast skill is very stable along the 8 days window. Please elaborate on the reasons behind this.

*The RMSAs are increasing over the course of the forecast period. However, we define*
*the "skill" as a ratio of RMSAs, that is with respect to the RMSA of the persistence.*

Sections 6.1 and 6.2: Please include the temporal (different cycles) component of the EOFs. These are indispensable for the interpretation of the results.

*We added the principal components to all EOFs shown in the manuscript. According to our understanding, however, our analysis represents EOFs of residuals with non-physical variance. As such the PCs do not yield a physical interpretation of the increments made to the forcing or the initial conditions respectively.*

Figure 3: Do you know the reason for the "sinusoidal" variation on the number of Aquarius observations?

*This is from the Aquarius orbit precession and missing data that is variable in time each day.*

Figure 11: Interesting how the assimilation can't reduce the RMSA or improve the CCA bellow 500m for the Argo and HOT data. But the same doesn't happen for the gliders data. Can you articulate about why this happens?

*The background error covariance is low below 500m, so it requires strong observational evidence to overcome the penalty in increments there. The sparse Argo provides an occasional profile; whereas, the SeaGliders (when available) provided many profiles per day in the same region.*

Figure 12: The standard deviation around the mean SS should be presented. Also, a grid would make the plot easier to read.

*The figure has been updated. The standard deviations and a grid have been added.*

Figures 13 and 14: Please include the time component of the EOFs.

*The figures have been updated including the principal components. Please see above*

[Figure]

*response on the non-physical variance of the PCs.*

**References**

G. Desroziers, O. Brachemi, and B. Hamadache. Estimation of the representativeness error caused by the incremental formulation of variational data assimilation. *Q.J.R. Meteorol. Soc.*, 127(575), 2001. doi: 10.1002/qj.49712757516.

G. Desroziers, L. Berre, B. Chapnik, and P. Poli. Diagnosis of observation, background and analysis‐error statistics in observation space. *Q.J.R. Meteorol. Soc.*, 131(613), 2005. doi: 10.1256/qj.05.108.

G. Desroziers, L. Berre, V. Chabot, and B. Chapnik. A posteriori diagnostics in an ensemble of perturbed analyses. *Monthly Weather Review*, 137(10), 2009. doi: 10.1175/2009MWR2778.1.

D. Matthews, B. S. Powell, and R. Milliff. Dominant spatial variability scales from observations around the Hawaiian islands. *Deep-Sea Research*, 58(10):979–987, 2011. doi: 10.1016/j.dsr. 2011.07.004.

D. Matthews, B. S. Powell, and I. Janeković. Analysis of four-dimensional variational state estimation of the Hawaiian waters. *Journal of Geophysical Research: Oceans*, 117:C03013, 2012. doi: 10.1029/2011JC007575.

BS Powell. Quantifying how observations inform a numerical reanalysis of hawaii. *Journal of Geophysical Research: Oceans*, 122(11):8427–8444, 2017.

J. M. A. C. Souza, B. S. Powell, A. C. Castillo-Trujillo, and P. Flament. The vorticity balance of the ocean surface in Hawaii from a regional reanalysis. *Journal of Physical Oceanography*, 45(2):424–440, 2015. doi: 10.1175/JPO-D-14-0074.1. URL http://dx.doi.org/10.1175/JPO-D-14-0074.1.

---

## Author Comment (AC3) · 7 Oct 2018

article

hyperref  graphics natbib

**Response to Reviewer #2**

*We thank the reviewer for the thoughtful and thorough review of the manuscript. We have made a major revision to the manuscript to address all of the reviewer's comments, and we believe that we have addressed the concerns raised. We accept all of the reviewer's comments.*

General comments: This paper presents a 10-year reanalysis of the Hawaiian Islands region, using IS4DVar and assimilating a variety of observations. The authors show that the state-estimates provide an improved representation of the (assimilated) observations compared to the forecasts, which is to be expected. It is unfortunate that they do not present comparison to any independent observations.

*The purpose of the I4D-Var method is to represent the observations by exploiting the linearized model dynamics. Therefor, all available observations are used to constrain this representation. The authors refrained from reserving data sets for validation as this is an arbitrary procedure that would result in less constrained state estimate. This is now clearly mentioned in the manuscript.*

The authors mention in the introduction and the summary that the system presented uses an updated model and data assimilation scheme, but do not elaborate as to how the model has been updated nor do they quantify the improvements. Mention of this "updated version" should be removed unless they are going to provide some quantitative comparison along with a description of the updates.

*The original text was rather vague in reference to "improvement", rather the suggestion is that we are conducting a decadal reanalysis of the Hawaiian region using the configuration as detailed in Souza et al. (2015). The test period used in Souza et al. (2015) was only an 18 month test case. This paper uses all available data over 10 years. The manuscript has been changed accordingly*

The authors present EOFs of the increment adjustment to the initial conditions and surface forcing, however physical interpretation of the increments is lacking.

*We added the principal components for the EOFs shown in the manuscript. We have revised the discussion of the EOFs. It must be kept in mind,however, that we are dealing with EOFs of residuals with non-physical variance. In the revised manuscript we discuss the spatial patterns of EOFs but clearly state that no physical modes came out of our EOF analysis.*

In general, the model and data assimilation methodology presented is sound, but the manuscript lacks insight that the authors can gain from their work. How do the results help to understand and improve their model and assimilation system? I recommend that the manuscript be returned to the authors for MINOR revision before acceptance

for publication in the Geoscientific Model Development Journal.

Specific comments: Line 13: remove the first HFR

*Corrected.*

Line 52: The HLCC is perhaps a weak current, but it is not a small current (in its spatial extent).

*Corrected.*

Figure 1: I wonder if it would be useful to label the islands, as you refer to them quite a bit in the text.

*Done.*

Lines 127-128: The inner loops are performed before the NL model is updated in the outer loop.

*This paragraph was re-written and the error was corrected.*

Lines 140-141: How close and how shallow?

*The text in the manuscript has been updated. It was specified that observations within on Rossby radius ($\sim$ 80 km) of the boundaries are ignored. The term "shallow" has been removed.*

Section 2.3: The author does not explain if super-observations are used and how the observation error variances are specified.

*For most datasets, we do not use "super-observations". In a least-squares minimization, finding the minimum variance of co-located values is the same as averaging them and assigning a larger variance. That said, for HFR, we did combine some due to the large number of observations; however, oftentimes, we have only one in a single grid cell. Regions where there are more than one HF radial in a single grid cell are those nearer to the HFR source. As such, the angles are very close, and we tested averaging the angle and radial value versus projecting them. Some of the details are found in Souza et al. (2015).*

Line 201: Call the Big Island Hawai'I instead for consistency throughout the paper.

*Corrected.*

Line 206: . . . around Hilo Bay (on the northeast of the island) . . .

*Corrected.*
Line 207: And what happens if it's $< 80\%$, is it still used?

*We have updated the manuscript explaining the HFR measurements from any return location that it missing more than 20 percent of its data over the 4-day assimilation period are ignored.*

Line 208 on: Do you grid in space to get super-observations?

*Please see response above. HFR observations are averaged in space if necessary.*

Figure 3: Hard to read the legend

*The quality of all figures including labels and legends has been improved.*

Figure 4, 6: Legends and text are too small to read

*The quality of all figures including labels and legends has been improved.*

Line 246: and have higher errors . . .?

*That's correct for SSS fields. Here the assumed noise level is 0.2 ppt. Furthermore, due to the loss of two SeaGliders in 2014, the SeaGlider data stopped in mid-2014 and only very few salt measurements were available thereafter (sporadic Argo data). We have updated the manuscript to better explain the small impact of salinity on the cost function reduction.*

Line 243-245: This statement is very vague.

*The shape of the cost function depends on the type and number of observations. When the observations change, also the cost function will change. This has now been*

*clarified in the revised manuscript.*

Line 305: . . . better forecasts than other methods that perturb the state at single times. These methods may better reduce. . ... , but can add . . .. . .

*The sentence has been rephrased.*

Section 4: Comparisons are only made against observations that are assimilated. This limitation needs to be made clear in the section introduction. Are there not any other observations of the region that could provide comparison to independent observations (e.g. ship-based CTD observations) ?

*Please see response above. The purpose of the I4D-Var method is to represent the observations by exploiting the linearized model dynamics. Therefor, all available observations are used to constrain this representation. This is now clearly explained in the manuscript*

[Figure]

Figure 9 and line 344: Do you mean the atmospheric model has inaccuracies in the representation of the heat fluxes? Or something else? Is this evident in the surface forcing increments discussed later?

*This mismatch is driven by a combination of three things: The lee side of Hawaiʻi is characterized by low cloud cover resulting in a relatively strong diurnal cycle of surface heat fluxes and hence SST. This diurnal cycle of surface heat fluxes is not fully represented in the atmospheric model. The surface layer of the ocean model, on the other hand, is too thick to simulate the impact of larger heat fluxes on SST to its full extent.*

Line 356 on: Did you look at how these adjustments extended beyond the radar coverage regions? Some snapshot examples might be nice to elucidate how the currents deviated in the forecast and how they were improved in the state-estimation (as well as how it looked beyond the coverage region). This may help add some physical context to this analysis.

*Figure 14 of the revised manuscript shows the EOF1 of adjustments applied to the near-surface currents. It should be emphasized that increments are made due to all variables considered by the I4D-Var scheme. The individual contribution by e.g. HFR to velocity adjustments remains elusive.*

Line 371-372: This sentence is unclear.

*We rephrased the sentence.*

Line 380: It's not much worse, I would say "of the same magnitude".

*Corrected.*

Line 385-387: Is this because the model background errors are low and the observation errors are relatively high below 500m, so the state-estimate makes little adjustment to salinity below 500m?

*Exactly. The background error covariance is low below 500 m, so it requires strong*

*observational evidence to overcome the penalty in increments there. The sparse Argo provides an occasional profile; whereas, the SeaGliders (when available) provided many profiles per day in the same region.*

Figure 12D: Interesting that the forecast skill degrades within the first 12 hours for the radials.

*The reason behind this decrease is given by the fact that the radials are dominated by the semi-diurnal both baroclinic and barotropic tides. Hence a reduction of the skill after 12 hours can be expected. This explanation has been added to the manuscript.*

Lines 431-433. By 'background values' do you mean the background standard deviations? It is surprising that the SSH is adjusted so little. Does the relatively large adjustments to the velocities suggest the system may be over fitting to the HF radar observations? Can you comment on the relative increments for the different variables as a percentage of the background standard deviations?

*The change is given as percentage of the initial conditions. This has been clarified in the manuscript. The larges adjustments to the initial conditions of the*

[Figure]

*velocity field are made far away from the HFR in the areas dominated by the shear between the NEC and the HLCC. This is not indicative of an overfitting to the HFR data.*

Line 435: Positive everywhere, so this suggests a bias that is being corrected. Can you comment on bias?

*The text has been clarified. The EOF1 has the same sign across the entire model domain. This is indicative of a bias correction that varies on the time scale of days. The total adjustment, however, is given by the product of the EOF1 and the PC1.*

Line 437: Do you mean horizontal temperature gradients? Can you explain how this dynamical characteristic relates to the higher increments in this region.

*The discussion of the EOFs has been revised.*

Line 446: increasing or decreasing (you don't show the temporal expansion function

which could have both negative and positive sign)

*The PCs are now shown in Figures 13-15.*

Figure 14 a) mode 1 is all negative. Again, can you talk about bias.

*The fact that the EOF1 has the same sign across the domain is discussed in the revised version of the manuscript. We interpret this feature as a bias correction.*

Lines 491-493: it is not clear if the improvement you are referring to is relative to the 'older' model version, or relative to the forecasts (which is the comparison that has been made throughout the paper).

*The paragraph in the summary has been revised.*

**References**

J. M. A. C. Souza, B. S. Powell, A. C. Castillo-Trujillo, and P. Flament. The vorticity balance of the ocean surface in Hawaii from a regional reanalysis. *Journal of Physical Oceanography*, 45(2):424–440, 2015. doi: 10.1175/JPO-D-14-0074.1. URL http://dx.doi.org/10.1175/JPO-D-14-0074.1.

---

## Author Response (AR2)

**Response to Topical Editor**

**The authors would like to thank the Editor and Topical Editor for their careful review of our revised manuscript. We addressed every request and suggestion made by the Topical Editor. Please see below.**

Topical Editor Decision: Publish subject to minor revisions (review by editor) (31 Oct 2018) by Steven Phipps.

Comments to the Author:
Dear authors,

Thank you for uploading a revised version of your manuscript. I am now happy to accept it for publication, subject to a few final technical corrections.

In regard to code and data availability, I remind you that manuscripts published in Geoscientific Model Development must ensure that the reader has access to sufficient code and data to be able to reproduce the results. Indeed, it is a mandatory requirement for publication in the journal that the reader must be able to access the EXACT source code that you used.

From our previous discussion of this subject, I was left with the understanding that you used a pure version of ROMS 3.6 in the current study. However, in your response to Referee #1, you state that:

"Co-Author Powell is a ROMS code contributor, and the ROMS 3.6 codebase was used with modifications for the HF radar assimilation and others mentioned in Souza et al. (2015)."

Can you please clarify whether or not these modifications are distributed as part of the ROMS codebase. If yes, then please expand the "Code and data availability" section to provide sufficient information that would allow the reader to locate the specific version (or sub-version etc.) of the model within the ROMS source code repository. If no, then you must make your implementation of the ROMS codebase available (either a full model distribution or, at a bare minimum, any routines that you modified or added).

**We uploaded the additional source code to the department's ftp server and added a note to the *Code Availability* section stating: "*The specific ROMS FORTRAN source for this package is under the MIT-license and is available at: ftp://ftp.soest.hawaii.edu/powellb/roms-gmd/roms-gmd.tar.gz*"**

Otherwise, please consider the following technical corrections:

Page 5, lines 133-135 and Page 6, line 138: Please correct the formatting of these references.

**The references have been corrected.**

Page 6, line 137: Insert "the" before "observation"?

**A "the" was added.**

Page 7, line 178: Remove "is" before "referred"?

**The "is" was removed.**

Page 8, line 219: mix → mixed.

**Corrected.**

Page 10, line 271: Aguarius → Aquarius.

**Corrected.**

Page 19, line 527: island $\rightarrow$ islands.

**Corrected.**

Steven Phipps
Handling Topical Editor

[revised manuscript text omitted]